# GRAPH2SEQ: GRAPH TO SEQUENCE LEARNING WITH ATTENTION-BASED NEURAL NETWORKS

## ABSTRACT

The celebrated *Sequence to Sequence learning (Seq2Seq)* technique and its numerous variants achieve excellent performance on many tasks. However, many machine learning tasks have inputs naturally represented as graphs; existing Seq2Seq models face a significant challenge in achieving accurate conversion from graph form to the appropriate sequence. To address this challenge, we introduce a general end-to-end graph-to-sequence neural encoder-decoder architecture that maps an input graph to a sequence of vectors and uses an attention-based LSTM method to decode the target sequence from these vectors. Our method first generates the node and graph embeddings using an improved graph-based neural network with a novel aggregation strategy to incorporate edge direction information in the node embeddings. We further introduce an attention mechanism that aligns node embeddings and the decoding sequence to better cope with large graphs. Experimental results on bAbI, Shortest Path, and Natural Language Generation tasks demonstrate that our model achieves state-of-the-art performance and significantly outperforms existing graph neural networks, Seq2Seq, and Tree2Seq models; using the proposed bi-directional node embedding aggregation strategy, the model can converge rapidly to the optimal performance.

## 1 INTRODUCTION

The celebrated *Sequence to Sequence learning (Seq2Seq)* technique and its numerous variants achieve excellent performance on many tasks such as Neural Machine Translation (Bahdanau et al., 2014; Gehring et al., 2017), Natural Language Generation (NLG) (Song et al., 2017) and Speech Recognition(Zhang et al., 2017). Most of the proposed Seq2Seq models can be viewed as a family of encoder-decoders (Sutskever et al., 2014; Cho et al., 2014; Bahdanau et al., 2014), where an encoder reads and encodes a source input in the form of sequences into a continuous vector representation of fixed dimension, and a decoder takes the encoded vectors and outputs a target sequence. Many other enhancements including Bidirectional Recurrent Neural Networks (Bi-RNN) (Schuster & Paliwal, 1997) or Bidirectional Long Short-Term Memory Networks (Bi-LSTM) (Graves & Schmidhuber, 2005) as encoder, and attention mechanism (Bahdanau et al., 2014; Luong et al., 2015), have been proposed to further improve its practical performance for general or domain-specific applications.

Despite their flexibility and expressive power, a significant limitation with the Seq2Seq models is that they can only be applied to problems whose inputs are represented as sequences. However, the sequences are probably the simplest structured data, and many important problems are best expressed with a more complex structure such as graphs that have more capacity to encode complicated pair-wise relationships in the data. For example, one task in NLG applications is to translate a graph-structured semantic representation such as Abstract Meaning Representation to a text expressing its meaning (Banarescu et al., 2013). In addition, path planning for a mobile robot (Hu & Yang, 2004) and path finding for question answering in bAbI task (Li et al., 2015) can also be cast as graph-to-sequence problems.

On the other hand, even if the raw inputs are originally expressed in a sequence form, it can still benefit from the enhanced inputs with additional information (to formulate graph inputs). For example, for semantic parsing tasks (text-to-AMR or text-to-SQL), they have been shown better performance by augmenting the original sentence sequences with other structural information such as dependency parsing trees (Pust et al., 2015). Intuitively, the ideal solution for graph-to-sequence tasks is to build

a more powerful encoder which is able to learn the input representation regardless of its inherent structure.

To cope with graph-to-sequence problems, a simple and straightforward approach is to directly convert more complex structured graph data into sequences (Iyer et al., 2016; Gómez-Bombarelli et al., 2016; Liu et al., 2017), and apply sequence models to the resulting sequences. However, the Seq2Seq model often fails to perform as well as hoped on these problems, in part because it inevitably suffers significant information loss due to the conversion of complex structured data into a sequence, especially when the input data is naturally represented as graphs. Recently, a line of research efforts have been devoted to incorporate additional information by extracting syntactic information such as the phrase structure of a source sentence (Tree2seq) (Eriguchi et al., 2016), by utilizing attention mechanisms for input sets (Set2seq)(Vinyals et al., 2015a), and by encoding sentences recursively as trees (Socher et al., 2010; Tai et al., 2015). Although these methods achieve promising results on certain classes of problems, most of the presented techniques largely depend on the underlying application and may not be able to generalize to a broad class of problems in a general way.

To address this issue, we propose Graph2Seq, a novel attention-based neural network architecture for graph-to-sequence learning. The Graph2Seq model follows the conventional encoder-decoder approach with two main components, a graph encoder and a sequence decoder. The proposed graph encoder aims to learn expressive node embeddings and then to reassemble them into the corresponding graph embeddings. To this end, inspired by a recent graph representation learning method (Hamilton et al., 2017a), we propose an inductive graph-based neural network to learn node embeddings from node attributes through aggregation of neighborhood information for directed and undirected graphs, which explores two distinct aggregators on each node to yield two representations that are concatenated to form the final node embedding. In addition, we further design an attention-based RNN sequence decoder that takes the graph embedding as its initial hidden state and outputs a target prediction by learning to align and translate jointly based on the context vectors associated with the corresponding nodes and all previous predictions. Our code and data are available at `https://github.com/anonymous/Graph2Seq`.

Graph2Seq is simple yet general and is highly extensible where its two building blocks, graph encoder and sequence decoder, can be replaced by other models such as Graph Convolutional (Attention) Networks (Kipf & Welling, 2016; Velickovic et al., 2017) or their extensions (Schlichtkrull et al., 2017), and LSTM (Hochreiter & Schmidhuber, 1997). We highlight three main contributions of this paper as follows:

- We propose a new attention-based neural networks paradigm to elegantly address graph-to-sequence learning problems that learns a mapping between graph-structured inputs to sequence outputs, which current Seq2Seq and Tree2Seq may be inadequate to handle.

- We propose a novel graph encoder to learn a bi-directional node embeddings for directed and undirected graphs with node attributes by employing various aggregation strategies, and to learn graph-level embedding by exploiting two different graph embedding techniques. Equally importantly, we present an attention mechanism to learn the alignments between nodes and sequence elements to better cope with large graphs.

- Experimental results show that our model achieves state-of-the-art performance on three recently introduced graph-to-sequence tasks and significantly outperforms existing graph neural networks, Seq2Seq, and Tree2Seq models.

## 2 RELATED WORK

Our model draws inspiration from the research fields of graph representation learning, neural networks on graphs, and neural encoder-decoder models.

**Graph Representation Learning.** Graph representation learning has been proven extremely useful for a broad range of the graph-based analysis and prediction tasks (Hamilton et al., 2017b; Goyal & Ferrara, 2017). The main goal for graph representation learning is to learn a mapping that embeds nodes as points in a low-dimensional vector space. These representation learning approaches can be roughly categorized into two classes including matrix factorization-based algorithms and

random-walk based methods. A line of research learn the embeddings of graph nodes through matrix factorization (Roweis & Saul, 2000; Belkin & Niyogi, 2002; Ahmed et al., 2013; Cao et al., 2015; Ou et al., 2016). These methods directly train embeddings for individual nodes of training and testing data jointly and thus inherently transductive. Another family of work is the use of random walk-based methods to learn low-dimensional embeddings of nodes by exploring neighborhood information for a single large-scale graph (Duran & Niepert, 2017; Hamilton et al., 2017a; Tang et al., 2015; Grover & Leskovec, 2016; Perozzi et al., 2014; Velickovic et al., 2017).

GraphSAGE (Hamilton et al., 2017a) is such a technique that learns node embeddings through aggregation from a node local neighborhood using node attributes or degrees for inductive learning, which has better capability to generate node embeddings for previously unseen data. Our graph encoder is an extension to GraphSAGE with two major distinctions. First, we non-trivially generalize it to cope with both directed and undirected graphs by splitting original node into forward nodes (a node directs to) and backward nodes (direct to a node) according to edge direction and applying two distinct aggregation functions to these types of nodes. Second, we exploit two different schemes (pooling-based and supernode-based) to reassemble the learned node embeddings to generate graph embedding, which is not studied in GraphSAGE. We show the advantages of our graph encoder over GraphSAGE in our experiments.

**Neural Networks on Graphs.** Over the past few years, there has been a surge of approaches that seek to learn the representations of graph nodes, or entire (sub)graphs, based on Graph Neural Networks (GNN) that extend well-known network architectures including RNN and CNN to graph data (Gori et al., 2005; Scarselli et al., 2009; Li et al., 2015; Bruna et al., 2013; Duvenaud et al., 2015; Niepert et al., 2016; Defferrard et al., 2016; Yang et al., 2016; Kipf & Welling, 2016; Chen et al., 2018). A line of research is the neural networks that operate on graphs as a form of RNN (Gori et al., 2005; Scarselli et al., 2009), and recently extended by Li et al. (Li et al., 2015) by introducing modern practices of RNN (using of GRU updates) in the original GNN framework. Another important stream of work that has recently drawn fast increasing interest is graph convolutional networks (GCN) built on spectral graph theory, introduced by Bruna et al. (2013) and then extended by Defferrard et al. (2016) with fast localized convolution. Most of these approaches cannot scale to large graphs, which is improved by using a localized first-order approximation of spectral graph convolution (Kipf & Welling, 2016) and further equipping with important sampling for deriving a fast GCN (Chen et al., 2018).

The closely relevant work to our graph encoder is GCN (Kipf & Welling, 2016), which is designed for semi-supervised learning in transductive setting that requires full graph Laplacian to be given during training and is typically applicable to a single large undirected graph. An extension of GCN can be shown to be mathematically related to one variant of our graph encoder on undirected graphs. We compare the difference between our graph encoder and GCN in our experiments. Another relevant work is gated graph sequence neural networks (GGS-NNs) (Li et al., 2015). Although it is also designed for outputting a sequence, it is essentially a prediction model that learns to predict a sequence embedded in graph while our approach is a generative model that learns a mapping between graph inputs and sequence outputs. A good analogy that can be drawn between our proposed Graph2Seq and GGS-NNs is the relationship between convolutional Seq2Seq and RNN.

**Neural Encoder-Decoder Models.** One of the most successful encoder-decoder architectures is the sequence to sequence learning (Sutskever et al., 2014; Cho et al., 2014; Bahdanau et al., 2014; Luong et al., 2015; Gehring et al., 2017), which are originally proposed for machine translation. Recently, the classical Seq2Seq model and its variants have been applied to several applications in which these models can perform mappings from objects to sequences, including mapping from an image to a sentence (Vinyals et al., 2015c), models for computation map from problem statements of a python program to their solutions (the answers to the program) (Zaremba & Sutskever, 2014), the traveling salesman problem for the set of points (Vinyals et al., 2015b) and deep generative model for molecules generation from existing known molecules in drug discovery. It is easy to see that the objects that are mapped to sequences in the listed examples are often naturally represented in graphs rather than sequences.

Recently, many research efforts and the key contributions have been made to address the limitations of Seq2Seq when dealing with more complex data, that leverage external information using specialized neural models attached to underlying targeted applications, including Tree2Seq (Eriguchi et al., 2016), Set2Seq (Vinyals et al., 2015a), Recursive Neural Networks (Socher et al., 2010), and Tree-

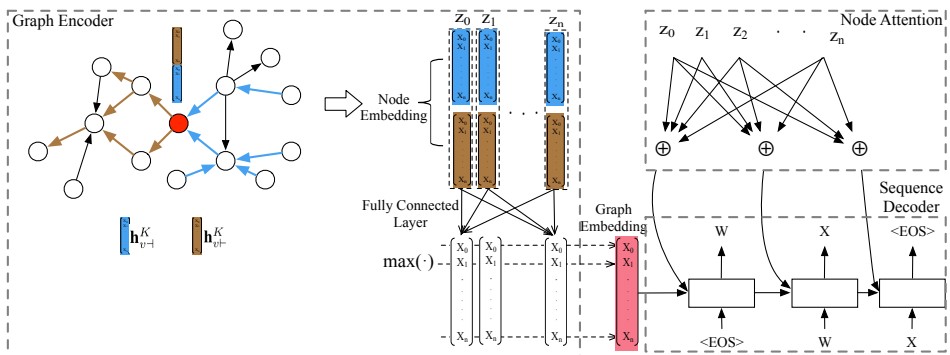

Figure 1: The framework of Graph2Seq model.

Structured LSTM (Tai et al., 2015). Due to more recent advances in graph representations and graph convolutional networks, a number of research has investigated to utilize various GNN to improve the performance over the Seq2Seq models in the domains of machine translation and graph generation (Bastings et al., 2017; Simonovsky & Komodakis, 2018; Li et al., 2018). There are several distinctions between these work and ours. First, our model is the first general-purpose encoder-decoder architecture for graph-to-sequence learning that is applicable to different applications while the aforementioned research has to utilize domain-specific information. Second, we design our own graph embedding techniques for our graph decoder while most of other work directly apply existing GNN to their problems.

## 3 GRAPH-TO-SEQUENCE MODEL

As shown in Figure 1, our graph-to-sequence model includes a graph encoder, a sequence decoder, and a node attention mechanism. Following the conventional encoder-decoder architecture, the graph encoder first generates node embeddings, and then constructs graph embeddings based on the learned node embeddings. Finally, the sequence decoder takes both the graph embeddings and node embeddings as input and employs attention over the node embeddings whilst generating sequences. In this section, we first introduce the node-embedding generation algorithm which derives the bi-directional node embeddings by aggregating information from both forward and backward neighborhoods of a node in a graph. Upon these node embeddings, we propose two methods for generating graph embeddings capturing the whole-graph information.

### 3.1 NODE EMBEDDING GENERATION

Inspired by Hamilton et al. (2017a), we design a new inductive node embedding algorithm that generates bi-directional node embeddings by aggregating information from a node local forward and backward neighborhood within $K$ hops for both directed and undirected graphs. In order to make it more clear, we take the embedding generation process for node $v \in \mathcal{V}$ as an example to explain our node embedding generation algorithm:[1]

1) We first transform node $v$'s text attribute to a feature vector, $\mathbf{a}_v$, by looking up the embedding matrix $\mathbf{W}_e$. Note that for some tasks where $v$'s text attribute may be a word sequence, one neural network layer, such as an LSTM layer, could be additionally used to generate $\mathbf{a}_v$.

2) We categorize the neighbors of $v$ into forward neighbors, $\mathcal{N}_{\vdash}(v)$, and backward neighbors, $\mathcal{N}_{\dashv}(v)$, according to the edge direction. In particular, $\mathcal{N}_{\vdash}(v)$ returns the nodes that $v$ directs to and $\mathcal{N}_{\dashv}(v)$ returns the nodes that direct to $v$;

3) We aggregate the **forward representations** of $v$'s forward neighbors $\{\mathbf{h}_{u\vdash}^{k-1}, \forall u \in \mathcal{N}_{\vdash}(v)\}$ into a single vector, $\mathbf{h}_{\mathcal{N}_{\vdash}(v)}^k$, where $k \in \{1, ..., K\}$ is the iteration index. In our experiments, we find that the aggregator choice, AGGREGATE$_k^{\vdash}$, may heavily affect the overall performance and we will discuss it later. Notice that at iteration $k$, this aggregator only uses the representations generated

---

[1]The pseudo-code of this algorithm can be found in the Appendix A.

at $k - 1$. The initial forward representation of each node is its feature vector calculated in step (1);

4) We concatenate $v$'s current forward representation, $\mathbf{h}_{v\vdash}^{k-1}$, with the newly generated neighborhood vector, $\mathbf{h}_{\mathcal{N}_\vdash(v)}^{k}$. This concatenated vector is fed into a fully connected layer with nonlinear activation function $\sigma$, which updates the forward representation of $v$, $\mathbf{h}_{v\vdash}^{k}$, to be used at the next iteration;

5) We update the **backward representation** of $v$, $\mathbf{h}_{v\dashv}^{k}$, using the similar procedure as introduced in step (3) and (4) except that operating on the backward representations instead of the forward representations;

6) We repeat steps (3)~(5) $K$ times, and the concatenation of the final forward and backward representation is used as the final bi-directional representation of $v$. Since the neighbor information from different hops may have different impact on the node embedding, we learn a distinct aggregator at each iteration.

**Aggregator Architectures.** Since a node neighbors have no natural ordering, the aggregator function should be invariant to permutations of its inputs, ensuring that our neural network model can be trained and applied to arbitrarily ordered node-neighborhood feature sets. In practice, we examined the following three aggregator functions:

**Mean aggregator:** This aggregator function takes the element-wise mean of the vectors in $\{\mathbf{h}_{u\vdash}^{k-1}, \forall u \in \mathcal{N}_\vdash(v)\}$ and $\{\mathbf{h}_{u\dashv}^{k-1}, \forall u \in \mathcal{N}_\dashv(v)\}$.

**LSTM aggregator:** Similar to (Hamilton et al., 2017a), we also examined a more complex aggregator based on an Long Short Term Memory (LSTM) architecture. Note that LSTMs are not inherently symmetric since they process their inputs sequentially. We use LSTMs to operate on unordered sets by simply applying them to a single random permutation of the node neighbors.

**Pooling aggregator:** In this aggregator, each neighbor's vector is fed through a fully-connected neural network, and an element-wise max-pooling operation is applied:

$$
\begin{aligned}
\text{AGGREGATE}_k^\vdash &= \max(\{\sigma(\mathbf{W}_{pool}\mathbf{h}_{u\vdash}^{k} + \mathbf{b}), u \in \mathcal{N}_\vdash(v)\}) \\
\text{AGGREGATE}_k^\dashv &= \max(\{\sigma(\mathbf{W}_{pool}\mathbf{h}_{u\dashv}^{k} + \mathbf{b}), u \in \mathcal{N}_\dashv(v)\})
\end{aligned}
\tag{1}
$$

where $\max$ denotes the element-wise max operator, and $\sigma$ is a nonlinear activation function. By applying max-pooling, the model can capture different information across the neighborhood set.

## 3.2 GRAPH EMBEDDING GENERATION

Most existing works of graph convolution neural networks focus more on node embeddings rather than graph embeddings since their focus is on the node-wise classification task. However, graph embeddings that convey the entire graph information are essential to the downstream decoder. In this work, we introduce two approaches (i.e., **_Pooling_**-based and **_Node_**-based) to generate these graph embeddings from the node embeddings.

**Pooling-based Graph Embedding.** In this approach, we investigated three pooling techniques: *max*-pooling, *min*-pooling and *average*-pooling. In our experiments, we fed the node embeddings to a fully-connected neural network and applied each pooling method element-wise. We found no significant performance difference across the three different pooling approaches; we thus adopt the *max*-pooling method as our default pooling approach.

**Node-based Graph Embedding.** In this approach, we add one **_super_** node, $v_s$, into the input graph, and all other nodes in the graph direct to $v_s$. We use the aforementioned node embedding generation algorithm to generate the embedding of $v_s$ by aggregating the embeddings of the neighbor nodes. The embedding of $v_s$ that captures the information of all nodes is regarded as the graph embedding.

## 3.3 ATTENTION BASED DECODER

The sequence decoder is a Recurrent Neural Network (RNN) that predicts the next token $y_i$, given all the previous words $y_{<i} = y_1, ..., y_{i-1}$, the RNN hidden state $s_i$ for time $i$, and a context vector $c_i$ that directs attention to the encoder side. In particular, the context vector $c_i$ depends on a set of node representations $(\mathbf{z}_1,...,\mathbf{z}_\mathcal{V})$ which the graph encoder maps the input graph to. Each node representation $\mathbf{z}_i$ contains information about the whole graph with a strong focus on the parts surrounding the

$i$-th node of the input graph. The context vector $c_i$ is computed as a weighted sum of these node representations and the weight $\alpha_{ij}$ of each node representation is computed by:

$$c_i = \sum_{j=1}^{\mathcal{V}} \alpha_{ij} h_j, \; where \; \alpha_{ij} = \frac{\exp(e_{ij})}{\sum_{k=1}^{\mathcal{V}} \exp(e_{ik})}, \; e_{ij} = a(s_{i-1}, h_j) \tag{2}$$

where $a$ is an *alignment model* which scores how well the input node around position $j$ and the output at position $i$ match. The score is based on the RNN hidden state $s_{i-1}$ and the $j$-th node representation of the input graph. We parameterize the alignment model $a$ as a feed-forward neural network which is jointly trained with other components of the proposed system. Our model is jointly trained to maximize the conditional log-probability of the correct description given a source graph. In the inference phase, we use the beam search to generate a sequence with the beam size = 5.

## 4 EXPERIMENTS

We conduct experiments to demonstrate the effectiveness and efficiency of the proposed method. Following the experimental settings in (Li et al., 2015), we firstly compare its performance with classical LSTM, GGS-NN, and GCN based methods on two selected tasks including bAbI Task 19 and the Shortest Path Task. We then compare Graph2Seq against other Seq2Seq based methods on a real-world application - Natural Language Generation Task. Note that the parameters of all baselines are set based on performance on the development set.

**Experimental Settings.** Our proposed model is trained using the Adam optimizer (Kingma & Ba, 2014), with mini-batch size 30. The learning rate is set to 0.001. We apply the dropout strategy (Srivastava et al., 2014) with a ratio of 0.5 at the decoder layer to avoid overfitting. Gradients are clipped when their norm is bigger than 20. For the graph encoder, the default hop size $K$ is set to 6, the size of node initial feature vector is set to 40, the non-linearity function $\sigma$ is ReLU (Glorot et al., 2011), the parameters of aggregators are randomly initialized. The decoder has 1 layer and hidden state size is 80. Since Graph2Seq with mean aggregator and pooling-based graph embeddings generally performs better than other configurations (we defer this discussion to Sec. 4.4), we use this setting as our default model in the following sections.

### 4.1 BABI TASK 19

**Setup.** The bAbI artificial intelligence (AI) tasks (Weston et al., 2015) are designed to test reasoning capabilities that an AI system possesses. Among these tasks, Task 19 (Path Finding) is arguably the most challenging task (see, e.g., (Sukhbaatar et al., 2015) which reports an accuracy of less than 20% for all methods that do not use strong supervision). We apply the transformation procedure introduced in (Li et al., 2015) to transform the description as a graph as shown in Figure 2. The left part shows an instance of bAbI task 19: given a set of sentences describing the relative geographical positions for a pair of objects $o_1$ and $o_2$, we aim to find the geographical path between $o_1$ and $o_2$. The question is then treated as finding the shortest path between two nodes, $N_{o_1}$ and $N_{o_2}$, which represent $o_1$ and $o_2$ in the graph. To tackle this problem with Graph2Seq, we annotate $N_{o_1}$ with text attribute *START* and $N_{o_2}$ with text attribute *END*. For other nodes, we assign their IDs in the graph as their text attributes. It is worth noting that, in our model, the START and END tokens are node features whose vector representations are first randomly initialized and then learned by the model later. In contrast, in GGS-NN, the vector representations of staring and end nodes are set as one-hot vectors, which is specially designed for the shortest path task.

To aggregate the edge information into the node embedding, for each edge, we additionally add a node representing this edge into the graph and assign the edge's text as its text attribute. We generate 1000 training examples, 1000 development examples and 1000 test examples where each example is a graph-path pair. We use a standard LSTM model (Hochreiter & Schmidhuber, 1997) and GGS-NN (Li et al., 2015) as our baselines. Since GCN (Kipf & Welling, 2016) itself cannot output a sequence, we also create a baseline that combines GCN with our sequence decoder.

**Results.** From Table 1, we can see that the LSTM model fails on this task while our model makes perfect predictions, which underlines the importance of the use of graph encoder to directly encode

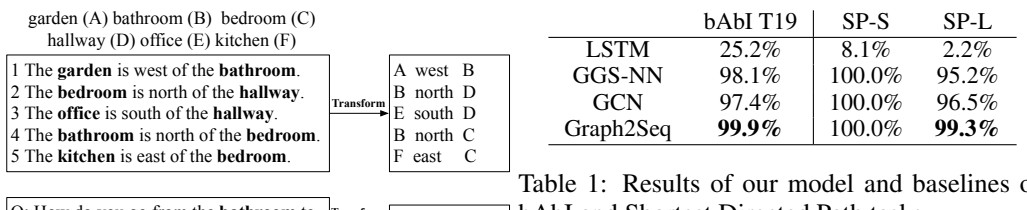

| | bAbI T19 | SP-S | SP-L |
|---|---|---|---|
| LSTM | 25.2% | 8.1% | 2.2% |
| GGS-NN | 98.1% | 100.0% | 95.2% |
| GCN | 97.4% | 100.0% | 96.5% |
| Graph2Seq | **99.9%** | 100.0% | **99.3%** |

Table 1: Results of our model and baselines on bAbI and Shortest Directed Path tasks.

Figure 2: Path Finding Example.

a graph instead of using sequence model on the converted inputs from a graph. Comparing to GGS-NN that uses carefully designed initial embeddings for different types of nodes such as START and END, our model uses a purely end-to-end approach which generates the initial node feature vectors based on random initialization of the embeddings for words in text attributes. However, we still significantly outperform GGS-NN, demonstrating the expressive power of our graph encoder that considers information flows in both forward and backward directions. We observe similar results when comparing our whole Graph2Seq model to GCN with our decoder, which mainly because the current form of GCN (Kipf & Welling, 2016) is designed for undirected graph and thus may have information loss when converting directed graph to undirected one as suggested in (Kipf & Welling, 2016).

## 4.2 SHORTEST PATH TASK

**Setup.** We further evaluate our model on the Shortest Path (SP) Task whose goal is to find the shortest directed path between two nodes in a graph, introduced in (Li et al., 2015). For this task, we created datasets by generating random graphs, and choosing pairs random nodes A and B which are connected by a unique shortest directed path. Since we can control the size of generated graphs, we can easily test the performance changes of each model when increasing the size of graphs as well. Two such datasets, SP-S and SP-L, were created, containing **S**mall (node size=5) and **L**arge graphs (node size=100), respectively. We restricted the length of the generated shortest paths for SP-S to be at least 2 and at least 4 for SP-L. For each dataset, we used 1000 training examples and 1000 development examples for parameter tuning, and evaluated on 1000 test examples. We choose the same baselines as introduced in the previous section.

**Results.** Table 1 shows that the LSTM model still fails on both of these two datasets. Our Graph2Seq model achieves comparable performance with GGS-NN that both models could achieve 100% accuracy on the SP-S dataset while achieves much better on larger graphs on the SP-L dataset. This is because our graph encoder is more expressive in learning the graph structural information with our dual-direction aggregators, which is the key to maintaining good performance when the graph size grows larger, while the performance of GGS-NN significantly degrades due to hardness of capturing the long-range dependence in a graph with large size. Compared to GCN, it achieves better performance than GGS-NN but still much lower than our Graph2Seq, in part because of both the poor effectiveness of graph encoder and incapability of handling with directed graph.

## 4.3 NATURAL LANGUAGE GENERATION TASK

**Setup.** We finally evaluate our model on a real-world application - Natural Language Generation (NLG) task where we translate a structured semantic representation—in this case a structured query language (SQL) query—to a natural language description expressing its meaning. As indicated in (Spiliopoulou & Hatzopoulos, 1992), the structure of SQL query is essentially a graph. Thus we naturally cast this task as an application of the graph-to-sequence model which takes a graph representing the semantic structure as input and outputs a sequence. Figure 3 illustrates the process of translation of an SQL query to a corresponding natural language description via our Graph2Seq model.[2]

We use the BLEU-4 score to evaluate our model on the WikiSQL dataset (Zhong et al., 2017), a corpus of 87,726 hand-annotated instances of natural language questions, SQL queries, and SQL

---

[2]The details of converting an SQL query into a graph is discussed in the Appendix B.

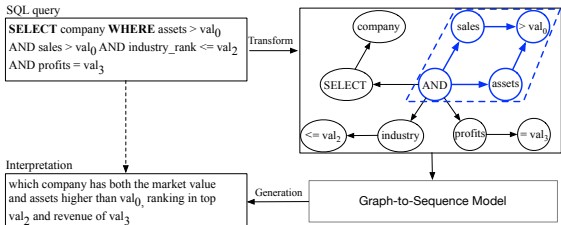

| | BLEU-4 |
|---|---|
| Seq2Seq | 20.91 |
| Seq2Seq + Copy | 24.12 |
| Tree2Seq | 26.67 |
| Graph2Seq-NGE | 34.28 |
| Graph2Seq-PGE | **38.97** |

Figure 3: A running example of the NLG task.

Table 2: Results on WikiSQL.

tables. WikiSQL was created as the benchmark dataset for the table-based question answering task (for which the state-of-the-art performance is 82.6% execution accuracy (Yu et al., 2018)); here we reverse the use of the dataset, treating the SQL query as the input and having the goal of generating the correct English question. These WikiSQL SQL queries are split into training, development and test sets, which contain 61297 queries, 9145 queries and 17284 queries, respectively.

Since the SQL-to-Text task can be cast as "machine translation" type of problems, we implemented several baselines to address this task. The first one is an attention-based sequence-to-sequence (Seq2Seq) model proposed by (Bahdanau et al., 2014); the second one additionally introduces the copy mechanism in the decoder side (Gu et al., 2016); the third one is a tree-to-sequence (Tree2Seq) model proposed by (Eriguchi et al., 2016) as our baseline. To apply these baselines, we convert an SQL query to a sequence or a tree using some templates which we discuss in detail in the Appendix.

**Results.** From Table 2, we can see that our Graph2Seq model performs significantly better than the Seq2Seq and Tree2Seq baselines. This result is expected since the structure of SQL query is essentially a graph despite its expressions in sequence and a graph encoder is able to capture much more information directly in graph. Tree2Seq achieves better performance compared to Seq2Seq since its tree-based encoder explicitly takes the syntactic structure of a SQL query into consideration. Two variants of the Graph2Seq models can substantially outperform Tree2Seq, which demonstrates that a general graph to sequence model that is independent of different structural information in complex data is very useful. Interestingly, we also observe that Graph2Seq-PGE (pooling-based graph embedding) performs better than Graph2Seq-NGE (node-based graph embedding). One potential reason is that the node-based graph embedding method artificially added a super node in graph which changes the original graph topology and brings unnecessary noise into the graph.

### 4.4 IMPACTS OF AGGREGATOR, HOP SIZE AND ATTENTION MECHANISM ON GARPH2SEQ MODEL

**Setup.** We now investigate the impact of the aggregator and the hop size on the Graph2Seq model. Following the previous SP task, we further create three synthetic datasets[3] : i) $\mathbf{SDP}_{DAG}$ whose graphs are directed acyclic graphs (DAGs); ii) $\mathbf{SDP}_{DCG}$ whose graphs are directed cyclic graphs (DCGs) that always contain cycles; iii) $\mathbf{SDP}_{SEQ}$ whose graphs are essentially sequential lines. For each dataset, we randomly generated 10000 graphs with the graph size 100 and split them as 8000/1000/1000 for the training/development/test set. For each graph, we generated an SDP query by choosing two random nodes with the constraints that there should be a unique shortest path connecting these two nodes, and that its length should be at least 4.

We create six variants of the Graph2Seq model coupling with different aggregation strategies in the node embedding generation. The first three (Graph2Seq-MA, -LA, -PA) use the **M**ean **A**ggregator, **L**STM **A**ggregator and **P**ooling **A**ggregator to aggregate node neighbor information, respectively. Unlike these three models that aggregate the information of both forward and backward nodes, the other two models (Graph2Seq-MA-F, -MA-B) only consider one-way information aggregating the information from the forward nodes or the information from the backward nodes with the mean aggregator, respectively. We use the path accuracy to evaluate these models. The hop size is set to 10.

**Impacts of the Aggregator.** Table 3 shows that on the $\mathrm{SDP}_{SEQ}$ dataset, both Graph2Seq-MA and Graph2Seq-PA achieve the best performance. On more complicated structured data, such as

---

[3]These datasets is valuable for other graph-based learning to test their performance regarding graph encoder and we will release them with our codes.

| Method | $\text{SDP}_{DAG}$ | $\text{SDP}_{DCG}$ | $\text{SDP}_{SEQ}$ |
|---|---|---|---|
| G2S-MA | **99.8%** | **99.2%** | **100%** |
| G2S-LA | 91.7% | 90.9% | 99.9% |
| G2S-PA | 96.7% | 98.4% | **100%** |
| G2S-MA-F | 78.8% | 98.7% | 70.2% |
| G2S-MA-B | 80.1% | 99.1% | 68.6% |

Table 3: Shortest path accuracy on three synthetic SDP datasets.

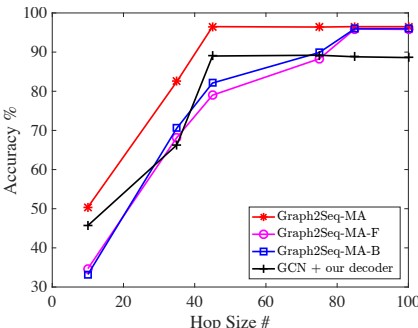

Figure 4: Test Results on $\text{SDP}_{1000}$.

$\text{SDP}_{DAG}$ and $\text{SDP}_{DCG}$, Graph2Seq-MA (our default model) also performs better than other variants. We can also see that Graph2Seq-MA performs better than Graph2Seq-MA-F and Graph2Seq-MA-B on $\text{SDP}_{DAG}$ and $\text{SDP}_{SEQ}$ since it captures more information from both directions to learn better node embeddings. However, Graph2Seq-MA-F and Graph2Seq-MA-B achieve comparable performance to Graph2Seq-MA on $\text{SDP}_{DCG}$. This is because in almost 95% of the graphs, 90% of the nodes could reach each other by traversing the graph for a given hop size, which dramatically restores its information loss.

**Impact of Hop Size.** To study the impact of the hop size, we create a $\text{SDP}_{DCG}$ dataset, $\text{SDP}_{1000}$ and results are shown in Figure 4. We see that the performance of all variants of Graph2Seq converges to its optimal performance when increasing the number of hop size. Specifically, Graph2Seq-MA achieves significantly better performance than its counterparts considering only one direction propagation, especially when the hop size is small. As the hop size increases, the performance differences diminish. This is the desired property since Graph2Seq-MA can use much smaller hop size (about the half) to achieve the same performance of Graph2Seq-MA-F or Graph2Seq-MA-B with a larger size. This is particularly useful for large graphs where increasing hop size may need considerable computing resources and long run-time. We also compare Graph2Seq with GCN, where the hop size means the number of layers in the settings of GCN. Surprisingly, even Graph2Seq-MA-F or Graph2Seq-MA-B can significantly outperform GCN with the same hope size despite its rough equivalence between these two architectures. It again illustrates the importance of the methods that could take into account both directed and undirected graphs. For additional experimental results on the impact of hop size for graphs of different sizes, please refer to the Table 4 in Appendix C.

**Impact of Attention Mechanism.** To investigate the impact of attention mechanism to the Graph2Seq model, we still evaluate our model on $\text{SDP}_{DAG}$, $\text{SDP}_{DCG}$ and $\text{SDP}_{SEQ}$ datasets but without considering the attention strategy. As shown in Table 4, we find that the attention strategy significantly improves the performance of all variants of Graph2Seq by at least 14.9%. This result is expected since for larger graphs it is more difficult for the encoder to compress all necessary information into a fixed-length vector; as intended, applying the attention mechanism in decoding enabled our proposed Graph2Seq model to successfully handle large graphs.

## 5 CONCLUSION

In this paper, we study the graph-to-sequence problem, introducing a new general and flexible Graph2Seq model that follows the encoder-decoder architecture. We showed that, using our proposed bi-directional node embedding aggregation strategy, the graph encoder could successfully learn representations for three representative classes of directed graph, i.e., directed acyclic graphs, directed cyclic graphs and sequence-styled graphs. Experimental results on three tasks demonstrate that our model significantly outperforms existing graph neural networks, Seq2Seq, and Tree2Seq baselines on both synthetic and real application datasets. We also showed that introducing an attention mechanism over node representation into the decoding substantially enhances the ability of our model to produce correct target sequences from large graphs. Since much symbolic data is represented as graphs and many tasks express their desired outputs as sequences, we expect Graph2Seq to be broadly applicable to unify symbolic AI and beyond.

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

## A    Pseudo-code of the Graph-to-sequence Algorithm

---

**Algorithm 1** Node embedding generation algorithm

---

**Input:** Graph $\mathcal{G}(\mathcal{V}, \mathcal{E})$; node initial feature vector $\mathbf{a}_v$, $\forall v \in \mathcal{V}$; hops $K$; weight matrices $\mathbf{W}^k$, $\forall k \in \{1, ..., K\}$; non-linearity $\sigma$; aggregator functions $\text{AGGREGATE}_k^{\vdash}$, $\text{AGGREGATE}_k^{\dashv}$, $\forall k \in \{1, ..., K\}$; neighborhood functions $\mathcal{N}_{\vdash}$, $\mathcal{N}_{\dashv}$

**Output:** Vector representations $\mathbf{z}_v$ for all $v \in \mathcal{V}$

1: $\mathbf{h}_{v\vdash}^0 \leftarrow \mathbf{a}_v$, $\forall v \in \mathcal{V}$
2: $\mathbf{h}_{v\dashv}^0 \leftarrow \mathbf{a}_v$, $\forall v \in \mathcal{V}$
3: **for all** $k = 1...K$ **do**
4:    **for all** $v \in \mathcal{V}$ **do**
5:        $\mathbf{h}_{\mathcal{N}_{\vdash}(v)}^k \leftarrow \text{AGGREGATE}_k^{\vdash}(\{\mathbf{h}_{u\vdash}^{k-1}, \forall u \in \mathcal{N}_{\vdash}(v)\})$
6:        $\mathbf{h}_{v\vdash}^k \leftarrow \sigma\left(\mathbf{W}^k \cdot \text{CONCAT}(\mathbf{h}_{v\vdash}^{k-1}, \mathbf{h}_{\mathcal{N}_{\vdash}(v)}^k)\right)$
7:        $\mathbf{h}_{\mathcal{N}_{\dashv}(v)}^k \leftarrow \text{AGGREGATE}_k^{\dashv}(\{\mathbf{h}_{u\dashv}^{k-1}, \forall u \in \mathcal{N}_{\dashv}(v)\})$
8:        $\mathbf{h}_{v\dashv}^k \leftarrow \sigma\left(\mathbf{W}^k \cdot \text{CONCAT}(\mathbf{h}_{v\dashv}^{k-1}, \mathbf{h}_{\mathcal{N}_{\dashv}(v)}^k)\right)$
9:    **end for**
10: **end for**
11: $\mathbf{z}_v \leftarrow \text{CONCAT}(\mathbf{h}_{v\vdash}^K, \mathbf{h}_{v\dashv}^K)$, $\forall v \in \mathcal{V}$

---

Algorithm 1 describes the embedding generation process where the entire graph $\mathcal{G} = (\mathcal{V}, \mathcal{E})$ and initial feature vectors for all nodes $\mathbf{a}_v$, $\forall v \in \mathcal{V}$, are provided as input. Here $k$ denotes the current hop in the outer loop. The $\mathbf{h}_{v\vdash}^k$ denotes node $v$'s forward representation which aggregates the information of nodes in $\mathcal{N}_{\vdash}(v)$. Similarly, the $\mathbf{h}_{v\dashv}^k$ denotes node $v$'s backward representation which is generated by aggregating the information of nodes in $\mathcal{N}_{\dashv}(v)$. Each step in the outer loop of Algorithm 1 proceeds as follows. First, each node $v \in \mathcal{V}$ in a graph aggregates the forward representations of the nodes in its immediate neighborhood, $\{\mathbf{h}_{u\vdash}^{k-1}, \forall u \in \mathcal{N}_{\vdash}(v)\}$, into a single vector, $\mathbf{h}_{\mathcal{N}_{\vdash}(v)}^k$ (line 5). Note that this aggregation step depends on the representations generated at the previous iteration of the outer loop, $k - 1$, and the $k = 0$ forward representations are defined as the input node feature vector. After aggregating the neighboring feature vectors, we concatenate the node current forward representation, $\mathbf{h}_{v\vdash}^{k-1}$, with the aggregated neighborhood vector, $\mathbf{h}_{\mathcal{N}_{\vdash}(v)}^k$. Then this concatenated vector is fed through a fully connected layer with nonlinear activation function $\sigma$, which updates the forward representation of the current node to be used at the next step of the algorithm (line 6). We apply similar process to generate the backward representations of the nodes (line 7, 8). Finally, the representation of each node $\mathbf{z}_v$ is the concatenation of the forward representation (i.e., $\mathbf{h}_{v\vdash}^K$) and the backward representation (i.e., $\mathbf{h}_{v\dashv}^K$) at the last iteration $K$.

## B    Structured Representation of the SQL Query

To apply Graph2Seq, Seq2Seq and Tree2Seq models on the natural language generation task, we need to convert the SQL query to a graph, sequence and tree, respectively. In this section, we describe these representations of the SQL query.

### B.1    Sequence Representation

We apply a simple template to construct the SQL query sequence: "SELECT + *<aggregation function>* + *<Split Symbol>* + *<selected column>* + WHERE + *<condition$_0$>* + *<Split Symbol>* + *<condition$_1$>* + ...".

### B.2    Tree Representation

We apply the SQL Parser tool[4] to convert an SQL query to a tree which is illustrated in Figure 5. Specifically, the root of this tree has two child nodes, namely SELECT LIST and WHERE CLAUSE. The child nodes of SELECT LIST node are the selected columns in the SQL query. The WHERE CLAUSE node has all occurred logical operators in the SQL query as its children. The children of a logical operator node are the columns on which this operator works.

---

[4]http://www.sqlparser.com

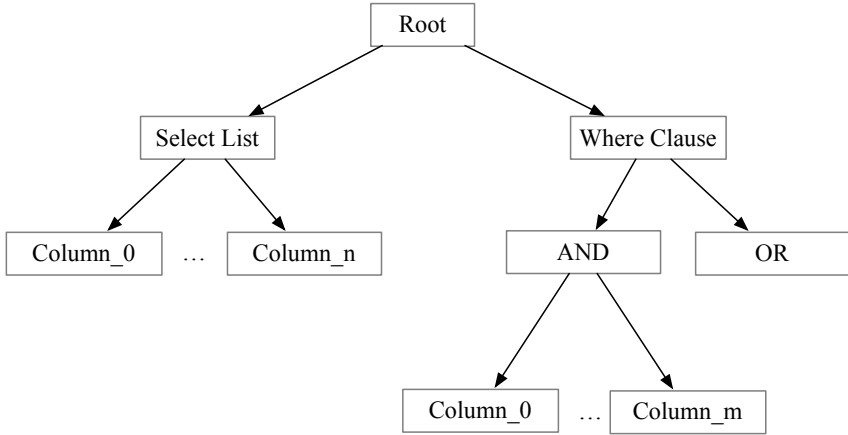

Figure 5: Tree representation of the SQL query.

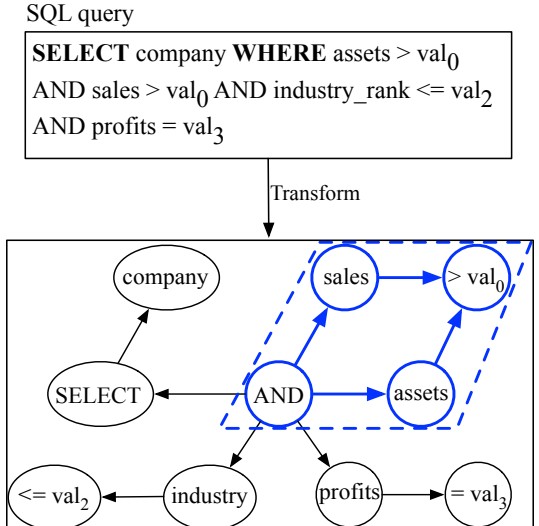

Figure 6: Graph representation of the SQL query.

### B.3  GRAPH REPRESENTATION

We use the following method to transform the SQL query to a graph:

**SELECT Clause.** For the SELECT clause such as "SELECT company", we first create a node assigned with text attribute *select*. This SELECT node connects with column nodes whose text attributes are the selected column names such as *company*. For the SQL queries that contain aggregation functions such as `count` or `max`, we add one aggregation node which is connected with the column node—their text attributes are the aggregation function names.

**WHERE Clause.** The WHERE clause usually contains more than one condition. For each condition, we use the same process as for the SELECT clause to create nodes. For example, in Figure 6, we create node *assets* and $>val_0$ for the first condition, the node *sales* and $>val_0$ for the second condition. We then integrate the constraint nodes that have the same text attribute (e.g., $>val_0$ in Figure 6). For a logical operator such as AND, OR and NOT, we create a node that connects with all column nodes that the operator works on (e.g., AND in Figure 6). These logical operator nodes then connect with SELECT node.

## C  MORE RESULTS ON THE IMPACT OF HOP SIZE

In Algorithm 1, we can see that there are three key factors in the node embedding generation. The first factor is the aggregator choice which determines how information from neighborhood nodes

| | | | SDP$_{100}$ | |
|---|---|---|---|---|
| Hop Size | Graph2Seq-MA-F | Graph2Seq-MA-B | Graph2Seq-MA | GCN (Kipf & Welling, 2016) + our decoder |
| 1 | 50.1% | 52.0% | 76.3% | 70.2% |
| 3 | 73.2% | 76.7% | 95.4% | 90.1% |
| 4 | 84.7% | 85.2% | 99.2% | 94.7% |
| 5 | 93.2% | 94.5% | 99.4% | 94.9% |
| 7 | 98.9% | 99.1% | 99.4% | 94.3% |
| 10 | 98.9% | 99.1% | 99.4% | 94.3% |
| | w/o attention | w/o attention | w/o attention | w/o attention |
| 10 | 85.8% | 86.3% | 89.6% | 83.1% |
| | | | SDP$_{1000}$ | |
| Hop Size | Graph2Seq-MA-F | Graph2Seq-MA-B | Graph2Seq-MA | GCN (Kipf & Welling, 2016) + our decoder |
| 10 | 34.7% | 33.2% | 50.4% | 45.7% |
| 35 | 68.2% | 70.6% | 82.5% | 66.3% |
| 45 | 79.0% | 82.1% | 96.5% | 89.0% |
| 75 | 88.3% | 89.9% | 96.4% | 89.2% |
| 85 | 95.9% | 96.0% | 96.5% | 88.8% |
| 100 | 95.8% | 96.0% | 96.5% | 88.6% |
| | w/o attention | w/o attention | w/o attention | w/o attention |
| 100 | 78.3% | 78.2% | 81.6% | 72.4% |

Table 4: Test Results on SDP$_{100}$ and SDP$_{1000}$.

is combined. The other two are the hop size ($K$) and the neighborhood function ($\mathcal{N}_\vdash(v)$, $\mathcal{N}_\dashv(v)$), which together determine which neighbor nodes should be aggregated to generate each node embedding. To study the impact of the hop size in our model, we create two SDP$_{DCG}$ datasets, SDP$_{100}$ and SDP$_{1000}$, where each graph has 100 nodes or 1000 nodes, respectively. Both of these two datasets contain 8000 training examples, 1000 dev examples and 1000 test examples. We evaluated three models, Graph2Seq-MA-F, Graph2Seq-MA-B and Graph2Seq-MA, on these two datasets; results are listed in Table 4.

We see that Graph2Seq-MA-F and Graph2Seq-MA-B could show significant performance improvements with increasing the hop size. Specifically, on the SDP$_{100}$ dataset, Graph2Seq-MA-F and Graph2Seq-MA-B achieve their best performance when the hop size reaches 7; further increases do not improve the overall performance. A similar situation is also observed on the SDP$_{1000}$ dataset; performance converges at the hop size of 85. Interestingly, the average diameters of the graphs in the two datasets are 6.8 and 80.2, respectively, suggesting that the ideal hop size for best Graph2Seq-MA-F performance should be the graph diameter. This should not be surprising; if the hop size equals the graph diameter, each node is guaranteed to aggregate the information of all reachable nodes on the graph within its embedding. Note that in the experiments on SDP$_{1000}$, in the $X$ ($X_{\dot{c}}10$) hop, we always use the aggregator in the *10*-th hop, because introducing too many aggregators (i.e., parameters) may make the model over-fitting.

Like Graph2Seq-MA-F, Graph2Seq-MA also benefited from increasing the hop size. However, on both datasets, Graph2Seq-MA could reach peak performance at a smaller hop size than Graph2Seq-MA-F. For example, on the SDP$_{100}$ dataset, Graph2Seq-MA achieves 99.2% accuracy once the hop size is greater than 4 while Graph2Seq-MA-F requires a hop size greater than 7 to achieve comparable accuracy; similar observations hold for the SDP$_{1000}$ dataset. Moreover, we can see that the minimum required hop size that Graph2Seq-MA could achieve its best performance is approximately the average radii (c.f. diameter) of the graphs, which are 3.4 and 40.1, respectively. Recall that the main difference between Graph2Seq-MA and Graph2Seq-MA-F (or Graph2Seq-MA-B) lies in whether the system aggregates information propagated from backward nodes; the performance difference indicates that by incorporating forward and backward nodes' information, it is possible for the model to achieve the best performance by traversing less of the graph. This is useful in practice, especially for large graphs where increasing hop size may consume considerable computing resources and run-time.

Table 4 also makes clear the utility of the attention strategy; the performance of both Graph2Seq-MA-F and Graph2Seq-MA decreases by at least 9.8% on SDP$_{100}$ and 14.9% on SDP$_{1000}$. This result is expected, since for larger graphs it is more difficult for the encoder to compress all necessary information into a fixed-length vector; as intended, applying the attention mechanism in decoding enabled our proposed Graph2Seq model to handle large graphs successfully.

As shown in Algorithm 1, the neighborhood function takes a given node as input and returns its directly connected neighbor nodes, which are then fed to the node embedding generator. Intuitively, to obtain a better representation of a node, this function should return all its neighbor nodes in the graph. However, this may result in high training times on large graphs. To address this, (Hamilton et al., 2017a) proposes a sampling method which randomly selects a fixed number of neighbor nodes from which to aggregate information at each hop. We use this sampling method to manage the neighbor node size at each aggregation step.

