# OpenReview forum: "Graph2Seq: Graph to Sequence Learning with Attention-Based Neural Networks"
_ICLR.cc/2019/Conference_

### Official Review · AnonReviewer2 · 2018-10-29
**Weak increment on graph to sequence tasks**

**Rating:** 4
**Confidence:** 5

**Review:**

The submission discusses a graph2seq architecture that combines a graph encoder that mixes GGNN and GCN components with an attentional sequence encoder. The resulting model is evaluated on three very simple tasks, showing small improvements over baselines.

I'm not entirely sure what the contribution of this paper is supposed to be. The technical novelty seems to be limited to new notation for existing work:
- (Sect. 3.1) The separation of forward/backward edges was already present in the (repeatedly cited) Li et al 2015 paper on GGNN (and in Schlichtkrull et al 2017 for GCN). The state update mechanism (a FC layer of the concatenation of old state / incoming messages) seems to be somewhere between a gated unit (as in GGNN) and the "add self-loops to all nodes" trick used in GCN; but no comparison is provided with these existing baselines.
- (Sect 3.2) The discussed graph aggregation mechanism are those proposed in Li et al and Gilmer et al; no comparison to these baselines is provided.
- (Sect. 3.3) This is a standard attention-based decoder; the fact that the memories come from a graph doesn't change anything fundamental.

The experiments are not very informative, as simple baselines already reach >95% accuracy on the chosen tasks. The most notable difference between GGS-NNs and this work seems to be the attention-based decoder, but that is not evaluated explicitly. For the rebuttal phase, I would like to ask the authors to provide the following:
- Experimental results for either GGS-NN with an attentional decoder, or their model without an attentional decoder, to check if the reported gains come from that. The final paragraph in Sect. 4 seems to indicate that the attention mechanism is the core enabler of the (small) experimental gains on the baselines.
- The results of the GCN/GG-NN models (i.e., just as an encoder) with their decoder on the NLG task.
- More precise definition of what they feel the contribution of this paper is, taking into account my comments from above.

Overall, I do not think that the paper in its current state merits publication at ICLR.

---

> ### Author Response · Authors · 2018-11-27
> **Response to Review #2: Part 1**
>
> We first would like to thank the referees for their very careful reading, for identifying subpar language, typos, and discrepancies in text, and for asking questions that will help us significantly improve the presentation.
>
> Q1: The submission discusses a graph2seq architecture that combines a graph encoder that mixes GGNN and GCN components with an attentional sequence encoder.
>
> Answer: our graph encoder is mainly inspired by the GraphSAGE work (Hamilton et al,. 2017a), which as we discussed its connection with other graph encoders such as GCN in related work. GGNN or GGS-NN is a relevant work but we are not sure which parts of our graph encoder are related to them. A good analogy between GGS-NN and GCN is the relationship between RNN(GRU) and CNN. In this spirit, our Graph2Seq model can be an analogy to convolutional Seq2Seq.
>
> Q2: The resulting model is evaluated on three very simple tasks, showing small improvements over baselines.
>
> Answer: there might be a misunderstanding about the experimental results. We respectfully argue two facts in terms of our experimental settings and results:
>
> 1) As we stated in the first paragraph of the Experiments, we chose the first two tasks to overlap with the tasks used by Li et al., (2015), which are synthetic datasets for better studying the characteristics of the proposed Graph2Seq model as well as other graph neural networks.
> 2) We chose the third dataset WikiSQL that is motivated by a real-world application - Natural Language Generation so that we can test the usefulness of our model as well as other baselines.
> 3) In Table 1, for bAbI T19 and SP-S tasks, our Graph2Seq achieve slightly better accuracy while for SP-L, our model significantly outperformed other graph neural networks based methods. In Table 2, for results on WikiSQL, our Graph2Seq models achieved much better results compared to other Seq2Seq and Tree2Seq models.
>
> Q3: (Sect. 3.1) The separation of forward/backward edges was already present in the (repeatedly cited) Li et al 2015 paper on GGNN (and in Schlichtkrull et al 2017 for GCN). The state update mechanism (a FC layer of the concatenation of old state / incoming messages) seems to be somewhere between a gated unit (as in GGNN) and the "add self-loops to all nodes" trick used in GCN; but no comparison is provided with these existing baselines.
>
> Answer: we agree with the reviewer that there are many ways to handle directed graphs like the one used in GGS-NN (Li et al., 2015) and syntactic GCN (Schlichtkrull et al., 2017). However, our design of combining the forward and backward representations is inspired by the bi-LSTM architecture, that is, generating the representation of a node for each direction (forward or backward), and then concatenating them. Intuitively, this architecture could explicitly represent the context of a node, i.e., the forward context representation and backward context representation. The state update mechanism of our graph encoder subsumes GCN as a special case, which is also discussed in (Hamilton et al,. 2017a).
> In addition, our bi-directional node embedding approach has been demonstrated and analyzed why it approaches the optimal performance faster than GCN and GGS-NN, as shown in Figure 4 and Table 1.
>
> Q4:  (Sect 3.2) The discussed graph aggregation mechanism are those proposed in Li et al and Gilmer et al; no comparison to these baselines is provided.
>
> Answer: we noticed that in (Li et al., 2015) and (Gilmer et al., 2017) they employed some soft attention based weighted node embedding to compute the graph-level embedding. This is very different from two graph embeddings we exploit in Sec 3.2. In particular, we have explored two graph embedding schemes:
> 1) Pooling-based graph embedding: we fed the node embeddings to a fully-connected neural network and then applied an pooling method (i.g. max-pooling, min-pooling, and average pooling) element-wise. This is different from weighted node embedding using soft attention.
> 2) Node-based graph embedding: we add on supernode into the input graph and all other nodes in the graph direct to this super node. Then the graph embedding can be obtained by aggregating the embeddings of the neighbor nodes. This approach has been discussed in the original GNN work (Scarselli et al., 2009) but with different aggregation approach.

---

> ### Author Response · Authors · 2018-11-27
> **Response to Review #2: Part 2**
>
> Q5: (Sect. 3.3) This is a standard attention-based decoder; the fact that the memories come from a graph doesn't change anything fundamental.
>
> Answer: Yes, this construction of attention is conceptually similar to the standard one. However, while we agree that our attention mechanism is *simple* once the construction is seen, we argue that it is not *trivial* to create this construction in the first place. If this were trivial, we might have expected earlier work such as (Li et al., 2015) and (Schlichtkrull et al., 2017) have already to use it, yet they did not. Therefore, we are the first one to design an attention mechanism between a sequence decoder and the graph node embeddings, which has also been demonstrated to be an important component for any Graph2Seq model with different graph encoder, as shown in Table 4 in Appendix C.
>
> Q6: The experiments are not very informative, as simple baselines already reach >95% accuracy on the chosen tasks.
>
> Answer: We assume the reviewer pointed to the results in Table 1. However, the simple baseline LSTM performed extremely poor in this task with only 25.2%, 8.1%, and 2.2% on bAbI T19, SP-S, and SP-L, respectively. Our approach and other state-of-the-art baselines such as GGS-NN and GCN can achieve the performance > 95%, highlighting the importance of considering a graph encoder for complex inputs like graphs. However, we can still see the large performance discrepancy when the graph size grows larger like SP-L, which challenges the effectiveness of existing graph encoders. The similar studies on the effect of the large graph are also provided in Figure 4.
>
> Q7: The most notable difference between GGS-NNs and this work seems to be the attention-based decoder, but that is not evaluated explicitly.
>
> Answer: Indeed, the attention mechanism plays an important role in our Graph2Seq model, which is not surprising since it has been widely recognized as a useful component in most of the encoder-decoder architectures nowadays. Thus, we just put the results in Table 4 for these “unsurprising results”. We refer the reviewer to Table 4 for more details in Appendix C.
>
> Compared to GGS-NN, other important difference is our bi-directional node embedding, which plays an important role for converging rapidly to the optimal performance compared to other graph encoders as shown in Figure 4.
>
> Q8: Experimental results for either GGS-NN with an attentional decoder, or their model without an attentional decoder, to check if the reported gains come from that. The final paragraph in Sect. 4 seems to indicate that the attention mechanism is the core enabler of the (small) experimental gains on the baselines.
>
> Answer: As we shown in Table 4, Attention mechanism is a core component for our graph encoder and GCN, which as discussed before is reasonable just like Seq2Seq with attention compared again vanilla Seq2Seq model.
>
> Q9: The results of the GCN/GG-NN models (i.e., just as an encoder) with their decoder on the NLG task.
>
> Answer: Although this is an interesting suggestion, this is slightly out of the scope of this paper. First of all, our Graph2Seq model is proposed to serve a generalized Seq2Seq model for graph inputs. On NLG task, we have already demonstrated the superior performance of Graph2Seq over Seq2Seq and Tree2Seq models. Second, in the previous two tasks (in Table 1), we also demonstrated the advantages of our Graph2Seq over GGS-NN and GCN. Finally, as we mentioned in the Introduction, “Graph2Seq is simple yet general and is highly extensible where its two building blocks, graph encoder and sequence decoder, can be replaced by other models”. We have released our code and data, and we would be happy to see more researchers and practitioners in using/adopting our Graph2Seq model for different tasks.

---

> > ### Comment · AnonReviewer2 · 2018-11-27
> > **Still no clarity on original contributions of the paper**
> >
> > Thank you for your clarifications. However, it remains unclear to me what the authors are claiming as the contribution of this paper. The introduction lists three contributions: (1) Attention-based graph-to-sequence learning; (2) a new bi-directional graph encoder with new graph embedding techniques; (3) experiments that show the value of these contributions.
> >
> > Regarding (1), I can list the following papers from memory (to avoid searching for something which may reveal the identity of the authors) that use graph2seq architectures:
> >  https://arxiv.org/abs/1704.04675 (EMNLP'17)
> >  https://arxiv.org/abs/1711.00740 (ICLR'18)
> >  http://aclweb.org/anthology/P18-1026 (ACL'18)
> >  https://openreview.net/forum?id=H1ersoRqtm (ICLR'19 submission)
> >  https://openreview.net/forum?id=B1fA3oActQ (ICLR'19 submission)
> >
> > Some of these works use attention over graph-generated embeddings (e.g., the first and oldest on the list), and I am sure that substantially more such works exist, as the idea is absolutely straightforward. Hence, I do not believe that you can claim novelty on the graph2seq structure, or the atttention mechanism over graph-generated node embeddings.
> >
> > Hence, the claim to the originality of (1) should be removed from the paper.
> >
> >
> > Now, if contribution (2), the graph encoder, is the remaining contribution of the paper, then there are many baselines from the literature, on better-known datasets, which the authors could compare their work to. This was not done (neither in the original submission nor in the revisions), and hence it remains unclear what the value of this contribution is. Indeed, the chosen baselines (GGS-NN and GCN) are the earliest representatives of (deep) graph message passing models, and more recent work on graph encoders from the last 3 years has been ignored.
> >
> > While the additional experiments in the second revision include valuable ablations, the lack of a comparison to current state of the art baselines makes it hard to judge if (2) indeed an improvement in the construction of graph encoders. The only information we obtain is from the new Table 4, which indicates that the new encoder beats GCN on a new synthetic dataset, but we don't know if this is due to the dataset, the fact that GCN is a weak baseline, or if the newly proposed encoder is actually better. All reviewers have asked for more informative experiments on this question, but the authors have declined to offer any results beyond this new Table 4. I believe that more information on this is crucial for the quality of this paper.
> >
> >
> > Overall, I continue to believe that the paper in its current form should not be accepted as neither the text nor the experimental results clearly articulate what conclusions the reader should take away: The graph2seq idea is not a new one (with or without attention); and the differences in the graph encoder are not compared sufficiently with existing graph encoders to allow any conclusions. However, I have slightly raised my rating to reflect the additional experiments.

---

> > > ### Author Response · Authors · 2018-11-28
> > > **Clarification on original contributions of the paper: Part I**
> > >
> > > We first thank the reviewer for the swift reply on our newly posted rebuttal. The reviewer #2 has a good memory to point out these references without searching for our paper topic. However, we would like to point out the Reviewer #2 is a little bit harsh on our submission for questioning the novelty of our paper. We are very surprised to see that there are two parallel ICLR submissions that reviewer #2 pointed out against our submission. We also want to remind the reviewer #2 that the second and third references you pointed out are actually contemporaneous works as ours (our work was posted to Arxiv even early than these very recently accepted papers). Despite these facts, we still want to explain in detail about the key differences between our Graph2Seq model and these references reviewer #2 mentioned.
> > >
> > >
> > > Q1: Regarding (1), I can list the following papers from memory (to avoid searching for something which may reveal the identity of the authors) that use graph2seq architectures:
> > >  https://arxiv.org/abs/1704.04675 (EMNLP'17)
> > >  https://arxiv.org/abs/1711.00740 (ICLR'18)
> > >  http://aclweb.org/anthology/P18-1026 (ACL'18)
> > >  https://openreview.net/forum?id=H1ersoRqtm (ICLR'19 submission)
> > >  https://openreview.net/forum?id=B1fA3oActQ (ICLR'19 submission)
> > >
> > > Some of these works use attention over graph-generated embeddings (e.g., the first and oldest on the list), and I am sure that substantially more such works exist, as the idea is absolutely straightforward. Hence, I do not believe that you can claim novelty on the graph2seq structure, or the attention mechanism over graph-generated node embeddings.  Hence, the claim to the originality of (1) should be removed from the paper.
> > >
> > > ---------------------------------------------------------------------------------------------------------------------------
> > > Answer: We will explain the differences between our proposed Graph2Seq model and existing works as follows. More importantly, why our Graph2Seq model is a novel general end-to-end neural network for learning the mapping between any graph inputs and sequence outputs independent of underlying applications.
> > >
> > > 1) https://arxiv.org/abs/1704.04675 (Bastings et al., EMNLP'17)
> > > We have already discussed the differences between our Graph2Seq model and Bastings's model in the response of Q3 for reviewer #1. In order to make it easy to follow, we have rephrased our previous arguments here:
> > >
> > > We noticed that Bastings et al., (2017) has utilized GCN for improving the encoder of the neural machine translation system, as we discussed in the related work. However, we would like to point out several major differences between their model and our Graph2Seq model in terms of model architecture:
> > >
> > > a) First of all, Bastings et al., (2017)  are not claiming that they developed general Graph-to-Sequence learning model. Instead, they claimed that "GCNs use predicted syntactic dependency trees of source sentences to produce representations of words (i.e. hidden states of the encoder) that are sensitive to their syntactic neighborhoods". In other words, they just used a version of GCNs to enhance sequence encoder to better capture syntactic information by taking into account syntactic dependency trees with the original sentence sequence SOLELY for neural machine translation. Therefore, many of their choices are centered around how to design a specific graph encoder for the original Seq2Seq model. These particular choices are listed in the subsequent items.
> > > b) Since Bastings’s encoder are built on GCN, which itself is derived from spectral graph convolutional neural networks, their model can be only used under transductive settings. In contrast, our graph encoder can be used under both transductive and inductive settings.
> > > c) Bastings's GCNs are on top of CNN or LSTM layers on input sentence, while our graph coder initializes all node embeddings as random vectors.
> > > d) Although Bastings’s encoder takes into account both incoming and outgoing edges as well as the edge labels, they only compute a single node embedding using the information from both directions. This is quite different from our bidirectional aggregation strategies, which is inspired from the bi-LSTM architecture, where we generate the representation of a node for each direction (forward or backward), and then concatenating them. Intuitively, this architecture could explicitly represent the context of a node, i.e., the forward context representation and backward context representation.
> > > e) Bastings et al. (2017) used the same attention based decoder of Bahdanau et al. (2015) while we design an attention-based decoder over the graph node embeddings. In other words, Bastings et al. (2017) used a domain knowledge to pre-select the attention applied only on original words in a sentence and completely ignore other nodes in the graph. Therefore, our attention mechanism is independent of underlying specific tasks and generally applicable to different tasks.

---

> > > ### Author Response · Authors · 2018-11-28
> > > **Clarification on original contributions of the paper: Part II**
> > >
> > > 2) https://arxiv.org/abs/1711.00740 (Allamanis et al., ICLR'18)
> > > Allamanis et al., (2018) presented an application of learning to represent programs with graphs using existing GGN-SS model in Li et al., (2015). However, we did not find any new model for Graph-to-Sequence Learning mentioned in this paper. We are not sure why the reviewer brought this recent work up to argue against the novelty of our Graph2Seq model. Instead, our Grpah2Seq model can actually be applied to this application as well.
> > >
> > > 3) http://aclweb.org/anthology/P18-1026 (Beck et al., ACL'18)
> > > We thank the reviewer for pointing out Beck et al., (2018) to us. In this work, Beck et al. proposed a similar model to the one proposed by Bastings et al. (2017). The main difference between these two works is that Beck et al., (2018) used a variant of GGN-SS proposed by Li et al., (2015) as graph encoder while Bastings et al. (2017) used a variant of GCN proposed by Kipf & Welling, (2016). As we discussed the difference between our Graph2Seq model and Bastings et al. (2017) above, the similar arguments of the differences between our Graph2Seq model and Beck et al., (2018) can also be discussed. So we refer the reviewer and other readers to the detailed differences above in 1).  In short, since both Bastings et al. (2017) and Beck et al., (2018) are proposed mainly for attacking neural machine translation problems, many of their model design choices are tailored for these tasks. In contrast, our Graph2Seq model is independent of underlying applications and thus a truly general end-to-end learning framework for graph-to-sequence problems.
> > >
> > > 4) https://openreview.net/forum?id=H1ersoRqtm (Structured Neural Summarization, ICLR'19 submission)
> > > This work presented an improved Seq2Seq model for neural summarization task by leveraging GNN-SS in Li et al., (2015). Since this is another NLP application, they also leverage Bi-LSTM to firstly obtain initial word representation and then feed them to GNN-SS. This work is very similar to the work proposed by Beck et al., (ACL'18) since both models used GNN-SS as the encoder and LSTM as the decoder. However, as we mentioned before, this model is very different from our Graph2Seq model in terms of both graph encoder and attention mechanism. All previous arguments about the differences above can also be applied here.
> > >
> > > 5) https://openreview.net/forum?id=B1fA3oActQ (GraphSeq2Seq: Graph-Sequence-to-Sequence for Neural Machine Translation, ICLR'19 submission)
> > > This work presented a GraphSeq2Seq model dedicated to neural machine translation. Similar to previous works in (Bastings et al. 2017) and (Beck et al., 2018), they utilize the dependency tree of the sentence sequence to leverage the model of Gildea et al., (2018). Differently, instead of following the order of Bi-LSTM-GNN for graph encoder, they choose the opposite order GNN-Bi-LSTM for graph encoder. It is easy to see that this is a very specific choice for graph encoder design dedicated to neural machine translation task. In contrast, our Graph2Seq model is very different from this work with respecting to both graph encoder and attention mechanism since we aim to design a Graph2Seq model that is application independent.
> > >
> > >
> > > Q2: the chosen baselines (GGS-NN and GCN) are the earliest representatives of (deep) graph message passing models, and more recent work on graph encoders from the last 3 years has been ignored.
> > >
> > > ---------------------------------------------------------------------------------------------------------------------------
> > > Answer: As we discussed different line of neural network on graphs in the Related Work Section, GGS-NN (which was proposed by  Li et al., (2015)) has been state-of-the-art model in the line of graph recurrent networks, and GCN (which was proposed by Kipf & Welling, (2016)) has been the standard model in the line of graph convolutional networks. There are some variants of these two models for some particular applications but we haven’t found the well recognized better models for both lines so far. From the above recent references the reviewer has pointed out, most of them are just simply adopted one of these models for their graph encoder, which might demonstrate their effectiveness over other models as well. Therefore, we think our chosen baselines GGS-NN and GCN are appropriate. Otherwise, we would be looking forward to hearing more state-of-the-art models that reviewer #2 think are better.

---

> > > ### Author Response · Authors · 2018-11-28
> > > **Clarification on original contributions of the paper: Part III**
> > >
> > > Q3: The only information we obtain is from the new Table 4, which indicates that the new encoder beats GCN on a new synthetic dataset, but we don't know if this is due to the dataset, the fact that GCN is a weak baseline, or if the newly proposed encoder is actually better. All reviewers have asked for more informative experiments on this question, but the authors have declined to offer any results beyond this new Table 4. I believe that more information on this is crucial for the quality of this paper.
> > >
> > > ---------------------------------------------------------------------------------------------------------------------------
> > > Answer: Although we think our results in Table 1, 4, and Figure 4 have enough information to show that our graph encoder indeed has more impressive power than other well-known graph encoder baselines GGS-NN and GCN, we respect the reviewer #2’s comments and have performed a new set of experiments on NLG task. We replaced our graph encoder with GCN or GGS-NN with other components fixed such as graph embedding scheme and attention mechanism. The experimental results are shown below and we will incorporate these results in the final version of the paper.
> > > We would like to emphasize that our Graph2Seq is a general learning framework, and is highly extensible where its two building blocks, graph encoder, and sequence decoder, can be replaced by other more advanced models.
> > >
> > >
> > > 		                   BLEU-4
> > >
> > > Seq2Seq                   20.91
> > > Seq2Seq + Copy      24.12
> > > Tree2Seq                  26.67
> > > GCN                           35.99
> > > GGS-NN                    35.53
> > > Graph2Seq-NGE      34.28
> > > Graph2Seq-PGE      38.97
> > >
> > >
> > > ---------------------------------------------------------------------------------------------------------------------------
> > > Final Remark:
> > >
> > > we hope our replies clarify the reviewer’ concerns and are helpful in making the final recommendation.

---

> > > > ### Comment · AnonReviewer2 · 2018-11-28
> > > > **On novelty of the graph2seq architecture**
> > > >
> > > > First, as this seems to have been lost on the authors of this submission: The VarNaming task from Allamanis et al. takes a graph and produces a sequence of tokens (that make up a variable name) and the model discussed by Allamanis et al. is explicitly refered to as an instance of a "graph2seq architecture" (this I why remembered the paper in this context actually).
> > > >
> > > > Second, let me clarify why I brought up very recent submissions as well: If the idea is widespread enough that other papers just use it as a part of another model, you cannot claim that the idea is novel anymore. Just because no one felt it necessary to explicitly claim that they have a "general graph2seq architecture" instead of a "specific graph2seq architecture" (whatever the difference there is supposed to be), you can not pretend that this idea doesn't already exist. The idea of a graph2seq work is an obvious extension of the seq2seq, seq2tree, tree2seq, etc. models in the literature and the list of papers I cited shows that plenty of other authors have already used this idea.
> > > >
> > > > Third, your model obviously differs from the models from the literature and in other papers (as you discuss in great detail), but not in the general architecture of a graph encoder and a sequence decoder. That is perfectly fine, and a useful contribution, but just as the authors of the first paper using a bidirectional RNN in a seq2seq setting could not claim that their paper is the first to introduce a seq2seq architecture, you cannot claim novelty on the graph2seq architecture. This claim in the paper is factually wrong and should be removed.
> > > >
> > > > If you truly believe that your novel contribution is the graph2seq architecture, then you should focus the paper on that and I will lower my score substantially again, as I believe that claim to be trivially wrong. [The AC may then judge if I'm mistaken and you are correct in claiming novelty]
> > > >
> > > > But because you describe other things in your paper that may be interesting, I believe it is important to consider these other claimed contributions in my review. The core differences to other graph encoder models that you have identified are (a) your message aggregation strategy, (b) the explicit split between forward and backward direction and (c) the computation of the graph embedding from node embeddings. [I continue to discount your claim of novelty on attention over all graph nodes instead of attention over a subset of graph nodes, because the latter is a more complex case than the former]

---

> > > > > ### Author Response · Authors · 2018-11-29
> > > > > **Clarification on novelty of the graph2seq architecture**
> > > > >
> > > > > Q1: The VarNaming task from Allamanis et al. takes a graph and produces a sequence of tokens (that make up a variable name) and the model discussed by Allamanis et al. is explicitly refered to as an instance of a "graph2seq architecture" (this I why remembered the paper in this context actually).
> > > > >
> > > > > Answer: Yes, it is another good motivated application that a Graph2Seq model could be utilized.
> > > > >
> > > > > Q2: Just because no one felt it necessary to explicitly claim that they have a "general graph2seq architecture" instead of a "specific graph2seq architecture" (whatever the difference there is supposed to be), you can not pretend that this idea doesn't already exist...
> > > > >
> > > > > Answer: We respectfully disagree with the reviewer's comments that there is no difference between "general Graph2Seq model" and "Specific Graph2Seq model". Just as we discussed in our paper about "Neural Encoder-Decoder Models." in the Related work, we quote these text here: "There are several distinctions between these work and ours. First, our model is the first general-purpose encoder-decoder architecture for graph-to-sequence learning that is applicable to different applications while the aforementioned research has to utilize domain-specific information. Second, we design our own graph embedding techniques for our graph decoder while most of other work directly apply existing GNN to their problems."
> > > > >
> > > > > The reason these previous works such as (Bastings et al., 2017) and other references you pointed out (as I have discussed in my previous responses) do not claim their models are general Graph2Seq model is simple. Their models are tailored to utilize the domain knowledge of these NLP applications and thus are not readily applicable to a broad range of other Graph-to-Sequence applications. In addition, the motivation of their works is also centered around these applications such as improving neural machine translation in (Bastings et al., 2017) and (Beck et al., ACL'18). This is fundamentally different from our motivation, that is to only consider a general problem consisting of the graph inputs and sequence outputs regardless of applications.
> > > > >
> > > > > We truly believe that it would be a merit to have a general Graph2Seq model instead of a specific one that is designed for the particular application. And such a general model should be a better fit for a machine learning conference such as ICLR, as we noticed that all these previous models are published in the venue of NLP conferences.
> > > > >
> > > > > Q3: That is perfectly fine, and a useful contribution, but just as the authors of the first paper using a bidirectional RNN in a seq2seq setting could not claim that their paper is the first to introduce a seq2seq architecture, you cannot claim novelty on the graph2seq architecture.
> > > > >
> > > > > Answer: It seems that there is a misunderstanding here. We NEVER claim our Graph2Seq model is the FIRST one or we are the FIRST to propose such a model. We quote the sentence for our first contribution in the Introduction section here: "We propose a new attention-based neural networks paradigm to elegantly address graph- to-sequence learning problems that learns a mapping between graph-structured inputs to sequence outputs, which current Seq2Seq and Tree2Seq may be inadequate to handle."
> > > > >
> > > > > We think it is reasonable to state that we proposed a new Graph2Seq model to address a Graph-to-Sequence problem.  In the second contribution, we specifically articulate what's are new components in our Graph2Seq model, which the reviewer seems to generally agree with as well.
> > > > >
> > > > > Q4: The core differences to other graph encoder models that you have identified are (a) your message aggregation strategy, (b) the explicit split between forward and backward direction and (c) the computation of the graph embedding from node embeddings. [I continue to discount your claim of novelty on attention overall graph nodes instead of attention over a subset of graph nodes, because the latter is a more complex case than the former]
> > > > >
> > > > > Answer: We thank the reviewer agreed with our contributions as you summarized here except the novelty of our attention mechanism. However, we haven't seen existing related references that used Graph2Seq models in NLP applications have used a sophisticated way to select a subset of graph nodes to apply attention. Instead, they just simply used domain knowledge to apply attentions directly on word nodes (in a word sequence) such as neural machine translation in (Bastings et al., 2017) and (Beck et al., ACL'18), which are just part of the all nodes in a graph. Compared to these existing works, we think our attention over all graph nodes looks simple but definitely more general because it is a simple yet effective way to apply attention without considering the domain knowledge of any underlying applications.

---

> > > > ### Comment · AnonReviewer2 · 2018-11-28
> > > > **On designing experimental evaluations**
> > > >
> > > > As discussed in my earlier messages here, I feel that the value of these contributions would require proper evaluation in comparison with existing baselines. My main problem here is that these contributions are of course not restricted to the graph2seq setting, but can be relevant to any setting in which a graph encoder could be used (just as you keep pointing out that you could switch out the graph encoder in your experiments). The focus on new tasks in the graph2seq setting thus makes it harder to compare to the rich literature on graph encoders. Let me discuss in detail what I see in the experiments, and what I would expect.
> > > >
> > > > On contribution (a), we have Table 3, which indicates diffferences between the three aggregation methods; however, this is not compared to the aggregation strategies from the literature (in the GGNN case, summation; in the GCN case, summation weighted by the renormalized adjacency matrix; in the GAT case, attention), no theoretical analysis is provided, and so the value of this contribution relative to the existing literature remains unclear. Table 3 also suffers from reporting results on a task on synthetic data where node labels play no role, and thus may not be indicative of a general result.
> > > >
> > > > On contribution (b), we have Table 3 and the new Table 4. These indicate that using only forward/backward information leads to worse results, but does not compare to the setting used in GGNN and R-GCN, in which different edge types are used for the forward/backward direction, allowing for different message passing functions. These functions can conceivably learn to use "half" of the hidden dimensions for forward and the rest for backwards information, and so you would expect them to perform similarly or better (as they can adapt to the importance of the directionality). Again, no experiments are provided to compare to these baselines and thus we cannot conclude anything. As for (a), the fact that the task is synthetic and largely label-agnostic additionally adds doubts about the generalizability of these results.
> > > >
> > > > Finally, on contribution (c), you have now provided additional information (and thank you for quickly providing these additional results!). I assume that your row on GGS-NN uses the weighted sum aggregation method from Li et al. 2015, but I'm not sure how you compute the graph-level representation for GCN. This is finally an experiment in which the effect of a modeling choice you made can be compared in isolation to the literature.
> > > >
> > > >
> > > > Overall, I am just disappointed in the design of your experiments: To evaluate the effect of different contributions, it would be crucial to (i) evaluate each contribution on its own, and keep everything else fixed, (ii) compare to existing work, not just new ablations of your own work, and (iii) use well-known tasks whereever possible. Your original submission was lacking on all three of these criteria, and your additional experiments are slowly moving towards fixing (i). To be constructive, these are the things that I would expect to see in a good evaluation of your paper:
> > > >  (1) Model variations using sum aggregation (as in GGNN), weighted sum aggregation (as in GCN), mean aggregation, (max)pooling in the node aggregation part, while keeping everything else the same. At least on the WikiSQL data (which at least is not synthetic), or even better, the tasks from Gilmer et al. or any other paper from the literature. This would shed light on the value of contribution (a).
> > > >  (2) Model variations using only forward, only backwards, bidirectional, or both directions with different edge types/separate weights (as in GGNN/R-GCN), while keeping everything else the same. Again, at least on WikiSQL, preferably on a task from the literature. This would shed light on contribution (b).
> > > >  (3) Model variations using a node-based graph representation, a pooling-based graph representation, or a weighted sum as Li et al. and Gilmer et al. use, while keeping everything else the same (i.e., add one more variation to your updated Table 4). This would shed light on contribution (c).
> > > >
> > > > I apologize for not posting these explicit instructions originally. I understand that we are past the end of the rebuttal period, but want to point out that the authors chose to post their response on the last day of the rebuttal period.

---

> > > > > ### Author Response · Authors · 2018-11-29
> > > > > **Clarification on designing experimental evaluations**
> > > > >
> > > > > Q1: As discussed in my earlier messages here... The focus on new tasks in the graph2seq setting thus makes it harder to compare to the rich literature on graph encoders.
> > > > >
> > > > > Answer: It seems like there is a misunderstanding about the selected datasets. As we mentioned in the Experimental section, "Following the experimental settings in (Li et al., 2015), we firstly compare its performance with classical LSTM, GGS-NN, and GCN based methods on two selected tasks including bAbI Task 19 and the Shortest Path Task. We then compare Graph2Seq against other Seq2Seq based methods on a real-world application - Natural Language Generation Task."
> > > > >
> > > > > We intentionally selected the datasets that have been used in the literature, especially from our most relevant baseline GGS-NN (Li et al., 2015). So, the first two tasks are well recognized in the literature. For the third task, this is our motivated application, which aims to develop an interpretable natural language interface system (by translating SQL-to-Text back to ordinary users). As the reviewer's request, we have performed another new set of experiments on this real-application and provided the results in our previous reply, which we copied here:
> > > > >
> > > > > BLEU-4
> > > > >
> > > > > Seq2Seq                   20.91
> > > > > Seq2Seq + Copy      24.12
> > > > > Tree2Seq                  26.67
> > > > > GCN                           35.99
> > > > > GGS-NN                    35.53
> > > > > Graph2Seq-NGE      34.28
> > > > > Graph2Seq-PGE      38.97
> > > > >
> > > > > We think these comparisons are very clear and informative to show the advantage of our Graph2Seq model compared to these baselines.
> > > > >
> > > > > Q2: On contribution (a), we have Table 3, which indicates diffferences between the three aggregation methods; however, this is not compared to the aggregation strategies from the literature (in the GGNN case, summation; in the GCN case, summation weighted by the renormalized adjacency matrix; in the GAT case, attention), no theoretical analysis is provided, and so the value of this contribution relative to the existing literature remains unclear.
> > > > >
> > > > > Answer: As the reviewer admitted that we have performed a comprehensive set of experiments to study the three aggregation methods we proposed, we have no idea why we need to evaluate other aggregation methods in our Graph2Seq model. If the reviewer's the purpose is to compare if our aggregation methods are better than these in other baselines, the above results on NLG task as well as Table 1 in the paper have already shed light on the effectiveness of our aggregation methods.
> > > > >
> > > > > Q3: Table 3 also suffers from reporting results on a task on synthetic data where node labels play no role, and thus may not be indicative of a general result.
> > > > >
> > > > > Answer: This is not true. In both tasks, the node labels information have been utilized, as we stated in the paper "Comparing to GGS- NN that uses carefully designed initial embeddings for different types of nodes such as START and END, our model uses a purely end-to-end approach which generates the initial node feature vectors based on random initialization of the embeddings for words in text attributes."
> > > > >
> > > > > Q3: On contribution (b), we have Table 3 and the new Table 4. These indicate that using only forward/backward information leads to worse results, but does not compare to the setting used in GGNN and R-GCN, in which different edge types are used for the forward/backward direction, allowing for different message passing functions.
> > > > >
> > > > > Answer: Our Table 1, Figure 4,  and new results on NLG, we have compared our Graph2Seq with GGNN and GCN. We hope the reviewer can find the merits of our bi-directional embeddings from these updated(existing) results.
> > > > >
> > > > > Q4: on contribution (c), you have now provided additional information (and thank you for quickly providing these additional results!). I assume that your row on GGS-NN uses the weighted sum aggregation method from Li et al. 2015, but I'm not sure how you compute the graph-level representation for GCN.
> > > > >
> > > > > Answer: For GCN, we used our PGE graph embedding method since it has been shown to perform better than NGE one.
> > > > >
> > > > > Q5: Overall, I am just disappointed in the design of your experiments: ....
> > > > >
> > > > > Answer: We feel that the reviewer is very thoughtful and might have not considered this is a conference paper that is supposed to focus on reporting a new idea or other new findings. What you required are really good but with all these results we can almost write a survey paper on the effect of the combinations of various node aggregation methods, graph aggregation methods, and so on for Graph encoder, Graph2Seq and any neural network on graphs in the literature.
> > > > >
> > > > > We respectfully ask the reviewer to consider this is a conference paper and we would be happy to incorporate your comprehensive experimental suggestions in our future work.

---

> > > > > > ### Comment · AnonReviewer2 · 2018-11-29
> > > > > > **Final conclusions**
> > > > > >
> > > > > > This discussion seems to be going nowhere. To summarize, I believe the paper should not be published at ICLR (or at any venue), for the following two reasons:
> > > > > > (1) The title and sentences such as "We propose a new attention-based neural networks paradigm to elegantly address graph- to-sequence learning problems" discount the fact that there is a substantial set of prior work handling graph-to-sequence problems in the same way as the authors propose. The "paradigm" is concisely described by
> > > > > >   (node_reprs, graph_repr) = graph_encoder(graph)
> > > > > >   seq_decoder(initial_state=graph_repr, memories=node_reprs)
> > > > > > That is present in a substantial number of existing papers, and so I believe the authors cannot claim this as a contribution. This overclaiming in itself is reason for me to reject the paper.
> > > > > >
> > > > > > (2) The experiments are insufficient to compare the actually novel aspects of this submission with the existing work properly. The new results provided by the authors show that one of their models (with split forward/backward information and their proposed message aggregation strategy, but with a different graph representation) performs less well than simple baselines. This is a great example of why disentangled experiments are required to judge the value of a set of smaller design choices, and would usually lead me to rate a paper as 5 (marginally below acceptance threshold)
> > > > > >
> > > > > > I do not think that repeating these points will improve anything and will thus not reply to any further comments unless significant new points are raised.

---

> > > > > > > ### Author Response · Authors · 2018-11-29
> > > > > > > **Final Response: Reply to Final conclusions**
> > > > > > >
> > > > > > > First, we are deeply grateful for the reviewer2's time and effort for providing a swift and insightful feedback on our responses in the previous several days. Such intensive interactions between the reviewers and the authors are such a beautiful thing that ICLR provides to researchers who are very responsible for their jobs.
> > > > > > >
> > > > > > > Second, we would like to provide our final explanations to reviewer's final arguments to conclude our reponses as well.
> > > > > > >
> > > > > > > Q1: The "paradigm" is concisely described by
> > > > > > >   (node_reprs, graph_repr) = graph_encoder(graph)
> > > > > > >   seq_decoder(initial_state=graph_repr, memories=node_reprs)
> > > > > > > That is present in a substantial number of existing papers, and so I believe the authors cannot claim this as a contribution.
> > > > > > >
> > > > > > > Answer: We are now getting the point of the reviewer. So, this is an inappropriate wording problem. We apologize for this misunderstanding and will rephrase this setence like "We propose a new general attention-based neural networks model to elegantly address graphto-sequence learning problems ...". We hope this can resolve the reviewer's concerns of overclaim.
> > > > > > >
> > > > > > > Q2:  one of their models (with split forward/backward information and their proposed message aggregation strategy, but with a different graph representation) performs less well than simple baselines. This is a great example of why disentangled experiments are required to judge the value of a set of smaller design choices, and would usually lead me to rate a paper as 5 (marginally below acceptance threshold)
> > > > > > >
> > > > > > > Answer: As we explained the setting before, on NLG task, the GCN is combined with our PGE scheme, which should only be compared with Graph2Seq + PGE for an apple-to-apple comparison. We have already explained in the paper why the Graph2Seq+NGE performed worse than Garph2Seq + PGE as follows "One potential reason is that the node-based graph embedding method artificially added a super node in graph which changes the original graph topology and brings unnecessary noise into the graph." And we have clearly said that we choose Graph2Seq + PGE as our default model for this reason, "Since Graph2Seq with mean aggregator and pooling-based graph embeddings generally performs better than other configurations (we defer this discussion to Sec. 4.4), we use this setting as our default model in the following sections."
> > > > > > >
> > > > > > > But we do understand that different Graph embedding schemes lead to different performance. For instance, on NLG task, the GGN-SS with their own graph embedding scheme performed slightly better than Graph2Seq+NGE but much worse than our default model Graph2Seq+PGE.
> > > > > > >
> > > > > > > Q3: To summarize, I believe the paper should not be published at ICLR (or at any venue), for the following two reasons.
> > > > > > >
> > > > > > > Answer: We have explained why we think both reasons that the reviewer listed can be easlily addressed.  We hope the reviewer will take these final responses into account to give your final recommendation.
> > > > > > >
> > > > > > > Thanks much again for your time and great review job!

---

### Official Review · AnonReviewer1 · 2018-11-03
**Interesting paper**

**Rating:** 6
**Confidence:** 4

**Review:**

This paper proposes a graph to sequence transducer consisting of a graph encoder and a RNN with attention decoder.

Strengths:
- Novel architecture for graph to sequence learning.
- Improved performance on synthetic transduction tasks and graph to text generation.
Weaknesses:
- Experiments could provide more insight into model architecture design and the strengths and weaknesses of the model on non-synthetic data.

Transduction with structured inputs such as graphs is still an under-explored area, so this paper makes a valuable contribution in that direction. Previous work has mostly focused on learning graph embeddings producing outputs. This paper extends the encoder proposed by Hamilton et al (2017a) by modelling edge direction through learning “forward” and “backward” representations of nodes. Node embeddings are pooled to a form a graph embedding to initialize the decoder, which is a standard RNN with attention over the node embeddings.

The model is relatively similar to the architecture proposed by Bastings et al (2017) that uses a graph convolutional encoder, although the details of the graph node embedding computation differs. Although this model is presented in a more general framework, that model also accounted for edge directionality (as well as edge labels, which this model do not support).

This paper does compare the proposed model with graph convolutional networks (GCNs) as encoder experimentally, finding that the proposed approach performs better on shortest directed path tasks. However the paper could make difference between these architectures clearer, and provide more insight into whether different graph encoder architectures might be more suited to graphs with different structural properties.

The model obtains strong performance on the somewhat artificial bAbI and Shortest path tasks, while the strongest result is probably that of strong improvement over the baselines in SQL to text generation. However, very little insight is provided into this result. It would be interesting to apply this model to established NLG tasks such as AMR to text generation.

Overall, this is an interesting paper, and I’d be fine with it being accepted. However, the modelling contribution is relatively limited and it feels like for this to be a really strong contribution more insight into the graph encoder design, or more applications to real tasks and insight into the model’s performance on these tasks is required.

Editing notes:
Hamilton et al 2017a and 2017c is the same paper.
In some cases the citation format is used incorrectly: when the citation form part of the sentence, the citation should be inline. E.g. (p3) introduced by (Bruna et al., 2013)  -> introduced by Bruna et al. (2013).

---

> ### Author Response · Authors · 2018-11-27
> **Response to Review #1: Interesting paper**
>
> We first would like to thank the referees for their very careful reading, for identifying subpar language, typos, and discrepancies in text, and for asking questions that will help us significantly improve the presentation such as motivation and organization.
>
>
> Q1: Novel architecture for graph to sequence learning… Transduction with structured inputs such as graphs is still an under-explored area, so this paper makes a valuable contribution in that direction...
>
> We are very grateful for the kind comments of reviewers #1, in particular for your recognition of the key contributions of the paper.
>
> Q2: Experiments could provide more insight into model architecture design and the strengths and weaknesses of the model on non-synthetic data.
>
> Answer: we fully agree with the reviewer’s comments (which is also brought up by reviewer #3). According to your and reviewer #3’s comments, we have completely rewrite the paragraph that describes the experimental result of the SQL-to-Text task. We have also added/modified quite a few of paragraphs to better provide more insights on dataset selections, the motivation of the comparisons, and various characteristics of our Graph2Seq in terms of hop size, aggregation strategies, attention mechanism, and input graph sizes.
>
> Q3: The model is relatively similar to the architecture proposed by Bastings et al (2017)
>
> Answer: We noticed that Bastings et al., (2017) has utilized GCN for improving the encoder of the neural machine translation system, as we discussed in the related work. However, we would like to point out three major differences between their model and our Graph2Seq model in terms of model architecture:
>
> 1) Since Bastings’s encoder are built on GCN, which itself is derived from spectral graph convolutional neural networks, their model can be only used under transductive settings. In contrast, our graph encoder can be used under both transductive and inductive settings.
>  2) Although Bastings’s encoder takes into account both incoming and outgoing edges as well as the edge labels, they only compute a single node embedding using the information from both directions. This is quite different from our bidirectional aggregation strategies, which is inspired from the bi-LSTM architecture, where we generate the representation of a node for each direction (forward or backward), and then concatenating them. Intuitively, this architecture could explicitly represent the context of a node, i.e., the forward context representation and backward context representation.
> 3) Bastings et al. (2017) used the same attention based decoder of Bahdanau et al. (2015) while we design an attention-based decoder over the graph node embeddings. Therefore, our attention mechanism is independent of underlying specific tasks and generally applicable to different tasks.
>
> Q4: However the paper could make difference between these architectures clearer, and provide more insight into whether different graph encoder architectures might be more suited to graphs with different structural properties.
>
> Answer: we thank for the reviewer’s suggestions and we have added some more discussions about the differences between different graph encoder. For different graph encoders, the most important advantages of our graph encoder lie in the fact that, when the graph size increases, all graph encoders started to perform poorer due to losing global information of a graph. However, our bi-directional node embeddings help converge rapidly to the optimal performance as shown in Figure 4.
>
> Q5:  However, very little insight is provided into this result. It would be interesting to apply this model to established NLG tasks such as AMR to text generation.
>
> Answer: please see our previous response for providing more insights into the SQL-to-Text task. Conceptually, the AMR-to-Text task would be similar to the SQL-to-Text task, which we would be happy to explore in the near future.
>
> Q6: Overall, this is an interesting paper, and I’d be fine with it being accepted. However, the modeling contribution is relatively limited and it feels like for this to be a really strong contribution more insight into the graph encoder design, or more applications to real tasks and insight into the model’s performance on these tasks is required.
>
> Answer: we hope that our previous responses have helped better explain the novelty of our Graph2Seq model architecture over existing works. According to your and reviewer #3’s comments, we have rephrased our key contributions of our model, that is, a novel graph encoder to learn a bi-directional node embedding for directed and undirected graphs with node attributes by employing various aggregation strategies, and to learn graph-level embedding by exploiting two different graph embedding techniques. In addition, to the best of knowledge, our attention mechanism to learn the alignments between nodes and sequence elements to better cope with larger graphs is also proposed for the first time.

---

> > ### Comment · AnonReviewer1 · 2018-12-03
> > **Reponse to rebuttal and discussion**
> >
> > I thank the authors for their extensive response to all the reviewers' comments. However, I have to agree with the other reviewers that the current version of the paper is most likely not strong enough to be accepted. To summarize the concerns that I share:
> >
> > - The paper cannot claim that its main contribution is to propose a general graph-to-seq framework as (a) multiple models falling under the framework already exists and (b) the paper does not make extensive enough comparisons between different graph-to-seq approaches.
> >
> > - As pointed out multiple times by the other reviewers, there is nothing novel about the attention mechanism used by the model, as it is just standard attention applied to a different encoder (to which the attention is agnostic).
> >
> > - The proposed graph encoder is novel, but experiments do not clearly disentangle the various design choices which differentiate it from other models. The SQL to Text generation results reported in a comment to reviewer 2 is a step in the right direction, but it is not sufficient to answer all questions about comparing the model to alternative architectures, as raised in detail by reviewer 2.

---

> > > ### Author Response · Authors · 2018-12-03
> > > **Response to "Reponse to rebuttal and discussion"**
> > >
> > > We are sorry to hear that the reviewer 1 did not feel that this is a strong enough submission. However, we would like to clarify some points as follows:
> > >
> > > Q1: - The paper cannot claim that its main contribution is to propose a general graph-to-seq framework as (a) multiple models falling under the framework already exists and (b) the paper does not make extensive enough comparisons between different graph-to-seq approaches.
> > >
> > > Answer: as we have clearly replied the reviewer 2 about this issue, this is an inappropriate word problem and we would like to correct it as "a new general Graph-to-Seq model", which we deeply believe it is true. For (a), we have acknowledged these works in related works in (Bastings et al., 2017) and (Beck et al., ACL'18) and discussed the differences. Due to their nature of specific designs, they are not easily applicable to other more general Graph2Seq problems (Shortest path and Babi task) except NLP applications like machine translation. For (b), in general, there are only two popular Graph-to-Seq models (GCN+RNN or GGS-NN+RNN). Both of them have been extensively compared in three of our tasks. We are not sure how many tasks we need to perform can be called "extensive enough".
> > >
> > > Q2: - As pointed out multiple times by the other reviewers, there is nothing novel about the attention mechanism used by the model, as it is just standard attention applied to a different encoder (to which the attention is agnostic).
> > >
> > > Answer: we agree our attention is *simple" but definitely not trivial. Otherwise, it should be applied before in early literature. For this reason, we count it as a new one.
> > >
> > > Q3: - The proposed graph encoder is novel, but experiments do not clearly disentangle the various design choices which differentiate it from other models. The SQL to Text generation results reported in a comment to reviewer 2 is a step in the right direction, but it is not sufficient to answer all questions about comparing the model to alternative architectures, as raised in detail by reviewer 2.
> > >
> > > Answer: We have discussed there are no much general "alternative architectures", expect two popular Graph-to-Seq models (GCN+RNN or GGS-NN+RNN). The most recent progress (or applications) - two parallel ICLR submissions that reviewer 2 pointed out earlier could also be counted into two categories. For GCN and GGS-NN,  we have extensive comparisons among our graph encoder and these two models on three tasks. In the future, we would like to apply our Grpah2Seq model to more real applications, however, we think this is truly not related to the qualification of the acceptance of a paper.

---

### Official Review · AnonReviewer3 · 2018-11-07
**Interesting work but lacking some organization**

**Rating:** 6
**Confidence:** 4

**Review:**

This work proposes an end-to-end graph encoder to sequence decoder model with an attention mechanism in between.
Pros (+) :
+ Overall, the paper provides a good first step towards flexible end-to-end graph-to-seq models.
+ Experiments show promising results for the model to be tested in further domains.
Cons (-) :
- The paper would benefit more motivation and organization.

Further details below (+ for pros / ~ for suggestions / - for cons):

The paper could benefit a little more motivation:
- Mentioning a few tasks in the introduction may not be enough. Explaining why these tasks are important may help. What is the greater problem the authors are trying to solve?
- Same thing in the experiments, not well motivated, why these three? What characteristics are the authors trying to analyze with each of these tasks?

Rephrase the novelty argument:
- The authors argue to present a “novel attention mechanism” but the attention mechanism used is not new (Bahdanau 2014 a & b). The fact that it is applied between a sequence decoder and graph node embeddings makes the paper interesting but maybe not novel.
~ The novelty added by this paper is the “bi-edge-direction“ aggregation technique with the exploration of various pooling techniques. This could be emphasized more.

Previous work:
~ The Related Work section could mention Graph Attention Networks (https://arxiv.org/abs/1710.10903) as an alternative to the node aggregation strategy.

Aggregation variations:
+ The exploration between the three aggregator architectures is well presented and well reported in experiments.
~ The two Graph Embedding methods are also well presented, however, I didn’t see them in experiments. Actually, it isn’t clear at all if these are even used since the decoder is attending over node embeddings, not graph embedding… Could benefit a little more explanation

Experiments:
+ Experiments show some improvement on the proposed tasks compared to a few baselines.
- The change of baselines between table 1 for the first two tasks and table 2 for the third task is not explained and thus confusing.
~ There are multiple references to the advantage of using “bi-directional” node embeddings, but it is not clear from the description of each task where the edge direction comes from. A better explanation of each task could help.

Results:
- Page 9, the “Impact of Attention Mechanism” is discussed but no experimental result is shown to support these claims.


Some editing notes:
(1) Page 1, in the intro, when saying “seq2seq are excellent for NMT, NLG, Speech Reco, and drug discovery”: this last example breaks the logical structure of the sentence because it has nothing to do with NLP.
(2) Page 1, in the intro, when saying that “<...> a network can only be applied to sequential inputs”: replace network by seq2seq models to be exact.
(3) Typo on page 3, in paragraph “Neural Networks on Graphs”, on 8th line “usig” -> “using”
(4) Page 3, in paragraph “Neural Networks on Graphs”, the following sentence: “An extension of GCN can be shown to be mathematically related to one variant of our graph encoder on undirected graphs.” is missing some information, like a reference, or a proof in Appendix, or something else…
(5) Page 9, the last section of the “Impact of Hop Size” paragraph talks about the impact of the attention strategy. This should be moved to the next paragraph which discusses attention.
(6) Some references are duplicates:
|_ Hamilton 2017 a & c
|_ Bahdanau 2014 a & b

---

> ### Author Response · Authors · 2018-11-27
> **Response to Review #3: Part 1**
>
> We first would like to thank the referees for their very careful reading, for identifying subpar language, typos, and discrepancies in text, and for asking questions that will help us significantly improve the presentation such as motivation and organization.
>
> We are very grateful for the kind comments of reviewers #3, in particular for your recognition of the key contributions of the paper.
>
>
> Q1: The paper could benefit a little more motivation: motivated applications and creteria to select datasets
>
> Answer: we agree with the reviewer that the motivation for the proposed Graph2Seq model could be further enhanced. We try to explain the motivation clearer as discussed below:
>
> 1) The most important motivation is that the celebrated Seq2Seq model demands the problems inputs are represented in sequences, which may potentially hurt the capability of a neural network model can learn from data.
>      a) On one hand, the input data may be naturally represented in a more complex form. As we discussed in Introduction (second paragraph),  AMR-to-text, SQL-to-text, and path planning in the mobile robot are good examples of such problems, where the inputs are essentially in a graph format instead of a sequence.
>      b) On the other hand, even if the raw inputs are originally expressed in a sequence form, it can still benefit from the enhanced inputs with additional information (to formulate graph inputs). For example, for semantic parsing tasks (text-to-AMR or text-to-SQL), they have been shown better performance by augmenting the original sentence sequences with other structural information such as dependency parsing trees (Pust et al., 2015).
>      c) For these aforementioned problems, we presented Graph2Seq model which can be also viewed as a generalized Seq2Seq model for graphs inputs.
>
> 2) In general, we chose these three different tasks to demonstrate the effectiveness of our end-to-end Graph2Seq model over other Seq2Seq models. We have two main criteria to choose the selected datasets (applications):
>      a) The first main criteria is to choose the tasks that have been used in the previous literature so it is easier to make a fair comparison on well-known datasets. As we stated in the first paragraph of the Experiments section, we chose the first two tasks to overlap with the tasks used by Li et al., (2015).
>      b) The second main criteria is to choose a real-world application so that we can test the usefulness of our model as well as other baselines. The same evaluation philosophy has also used in (Li et al., 2015).
>      c) We also provided the ablation study of our model such as the impacts of aggregator and hop size on Graph2Seq model in Sec 4.4, where these important characteristics of our model were studied.
>      d) We agree with the reviewer’s comments that it would be more beneficial to briefly discuss the purpose or some emphasis of each task. Therefore, we have correspondingly revised the draft based on these comments.
>
> Q2: Rephrase the novelty argument “novel attention mechanism”
>
> Answer: we thank the reviewer for a very careful reading of the proper wording of “novel attention mechanism”. Indeed, the attention mechanism has been proposed by Bahdanau et al., (2014). However, what’s new in our setting is that we are the first one to design an attention mechanism between a sequence decoder and the graph node embeddings. While we agree that our attention mechanism is *simple* once the construction is seen, we argue that it is not *trivial* to create this construction in the first place. If this were trivial, we might have expected earlier work such as (Li et al., 2015) and (Schlichtkrull et al., 2017) have already to use it, yet they did not. We will revise it and make it more clear in the revision.
>
> Q3: The novelty added by this paper is the “bi-edge-direction“ aggregation technique with the exploration of various pooling techniques. This could be emphasized more.
>
> Answer: This is an accurate summary. We are grateful for your recognition of one of the key contributions of the paper. We will emphasize these points more in the revision.
>
> Q4: The Related Work section could mention Graph Attention Networks (https://arxiv.org/abs/1710.10903) as an alternative to the node aggregation strategy.
>
> Answer: Yes, we fully agree with you that the self-attention scheme proposed in the work of graph attention networks can be combined with our graph encoder as well. As we mentioned in the Introduction, “Graph2Seq is simple yet general and is highly extensible where its two building blocks, graph encoder, and sequence decoder, can be replaced by other models”.

---

> ### Author Response · Authors · 2018-11-27
> **Response to Review #3: Part 2**
>
> Q5: The two Graph Embedding methods are also well presented, however, I didn’t see them in experiments. Actually, it isn’t clear at all if these are even used since the decoder is attending over node embeddings, not graph embedding… Could benefit a little more explanation
>
> Answer: We indeed performed the experiments comparing these two graph embedding methods. We have included the additional experimental results in Table 2. It is worth noting that the graph embedding methods play an important role in the final performance as shown in Table 2. The graph-level embedding is computed as the initial input for our sequence decoder. The main reason for the significant performance discrepancy of these two graph embedding methods is that the super-node based graph embedding method artificially added a super node in the graph which changes the original graph topology and brings unnecessary noise into the graph. We have made it more clear in the updated submission.
>
> Q6: The change of baselines between table 1 for the first two tasks and table 2 for the third task is not explained and thus confusing.
>
> Answer: It appears that there is a misunderstanding. We realize now that our presentation obscured some important facets of the experimental settings of this paper.
> 1) For the third task, we changed the baselines to Seq2Seq based methods because the SQL-to-Text problem can be viewed as “machine translation” task which Seq2Seq is a state-of-the-art method for this problem. That is said, the previous baselines like LSTM, GGS-NN, and GCN cannot deal with problems well.
> 2) We have already shown in the previous tasks that Graph2Seq outperforms or matches LSTM, GGS-NN and GCN so here we just reported Graph2Seq to compare with other Seq2Seq based methods.
>
> Q7: Better explanation of “bi-directional” node embeddings
>
> Answer: Our design of combining the forward and backward representations is inspired by the bi-LSTM architecture, that is, generating the representation of a node for each direction (forward or backward), and then concatenating them. Intuitively, this architecture could explicitly represent the context of a node, i.e., the forward context representation and backward context representation. We think this design is reasonable and could be better at capturing the graph structure. In each task, the graph is either a directed or undirected graph. In both cases combining the node embedding from both contexts of a node are indeed beneficial as shown in their results.
>
> Q8: “Impact of Attention Mechanism”
>
> Answer: Attention-based decoder is widely recognized as a useful component in most of the encoder-decoder architectures nowadays. Thus, we just put the results in Table 4 for these “unsurprising results”. We refer the reviewer to Table 4 for more details in Appendix C.
>
> Q9: Other minor notes.
>
> Answer: we appreciated your careful reading and have fixed all of them based on your comments.

---

### Meta-Review · Area_Chair1 · 2018-12-11
**Interesting ideas, but a more targeted evaluation is needed**

**Confidence:** 4
**Recommendation:** Reject

**Metareview:**

Strengths:
The work proposes a novel architecture for graph to sequence learning.
The paper shows improved performance on synthetic transduction tasks and for graph to text generation.

Weaknesses:
Multiple reviewers felt that the experiments were insufficient to evaluate the novel aspects of the submission relative to prior work.  Newer experiments with the proposed aggregation strategy and a different graph representation were not as promising with respect to simple baselines.

Points of contention:
The discussion with the authors and one of the reviewers was particular contentious.
The title of the paper & sentences within the paper such as "We propose a new attention-based neural networks paradigm to elegantly address graph- to-sequence learning problems" caused significant contention, as this was perceived to discount the importance of prior work on graph-to-sequence problems which led to a perception of the paper "overclaiming" novelty.

Consensus:
Consensus was not reached, but both the reviewer with the lowest score and one of the reviewers giving a 6 came to the consensus that the experimental evaluation does not yet evaluate the novel aspects of the submission thoroughly enough.

Due to the aggregate score, factors discussed above (and others) the AC recommends rejection; however, this work shows promise and additional experimental work should allow a new set of reviewers to better understand the behaviour and utility of the proposed method.